# Constraining Generative Models for Engineering Design with Negative Data

**Lyle Regenwetter**                                           *regenwet@mit.edu*
*Massachusetts Institute of Technology*

**Giorgio Giannone**[*]                                        *ggiorgio@mit.edu*
*Amazon*
*Massachusetts Institute of Technology*

**Akash Srivastava**                                    *akash.srivastava8@ibm.com*
*MIT-IBM Watson AI Lab*

**Dan Gutfreund**                                              *dgutfre@us.ibm.com*
*MIT-IBM Watson AI Lab*

**Faez Ahmed**                                                      *faez@mit.edu*
*Massachusetts Institute of Technology*

**Reviewed on OpenReview:** *https://openreview.net/forum?id=FNBv2vweBI*

## Abstract

Generative models have recently achieved remarkable success and widespread adoption in society, yet they often struggle to generate realistic and accurate outputs. This challenge extends beyond language and vision into fields like engineering design, where safety-critical engineering standards and non-negotiable physical laws tightly constrain what outputs are considered acceptable. In this work, we introduce a novel training method to guide a generative model toward constraint-satisfying outputs using 'negative data' – examples of what to avoid. Our negative-data generative model (NDGM) formulation easily outperforms classic models, generating 1/6 as many constraint-violating samples using 1/8 as much data in certain problems. It also consistently outperforms other baselines, achieving a balance between constraint satisfaction and distributional similarity that is unsurpassed by any other model in 12 of the 14 problems tested. This widespread superiority is rigorously demonstrated across numerous synthetic tests and real engineering problems, such as ship hull synthesis with hydrodynamic constraints and vehicle design with impact safety constraints. Our benchmarks showcase both the best-in-class performance of our new NDGM formulation and the overall dominance of NDGMs versus classic generative models. We publicly release the code and benchmarks at https://github.com/Lyleregenwetter/NDGMs.

## 1 Introduction

Generative models have demonstrated impressive results in vision, language, and speech. However, even with massive datasets, they struggle with precision, often generating physically impossible images or factually incorrect text responses. These mistakes are examples of constraint violation; ideally, generative models would be constrained to only generate valid 'correct' samples. While constraint violation is a nuisance in image or text synthesis, it is a paramount concern in domains like engineering design with high-stakes (including safety-critical) constraints. Generative models synthesizing designs for car or airplane components, for example, may be subject to geometric restrictions (such as colliding components), functional requirements (such as load-bearing capacity), industry standards, and manufacturing limitations.

---

[*]Work completed at Massachusetts Institute of Technology

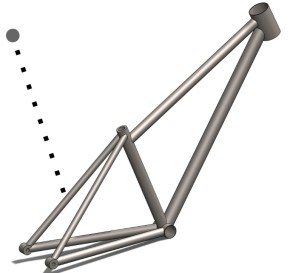
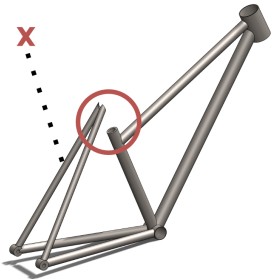
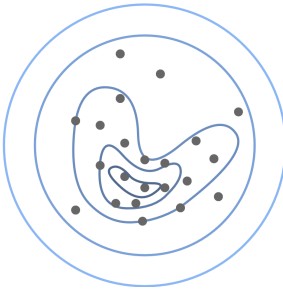
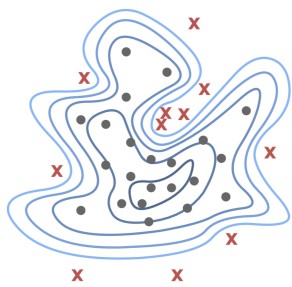

**(a)** Positive datapoints are constraint-satisfying examples that generative models should broadly replicate.

**(b)** Negative datapoints are constraint-violating examples that generative models should avoid.

**(c)** Generative models are typically trained using only positive data, causing them to ignore constraints.

**(d)** Negative data allows generative models to better satisfy constraints through tighter density estimates.

**Figure 1:** Negative data helps generative models learn real-world data distributions, which often have gaps in their support caused by constraints. For example, by examining bike frames with disconnected components, a model can better learn to generate geometrically valid frames.

As generative models are increasingly applied to engineering problems, their blatant violation of objective, ubiquitous, and non-negotiable constraints becomes increasingly problematic. We assert that this constraint-violation issue is largely attributable to the fact that generative models are classically shown only 'positive' (constraint-satisfying) datapoints during training, and are never exposed to 'negative' (constraint-violating) datapoints to avoid. Completely satisfying constraints using this training approach is equivalent to learning a binary classification problem with only one class present in the data, a challenging task. Instead, by studying negative data in addition to positive data, generative models can better avoid constraint-violating samples during generation (Figure 1). This aligns with their distribution-matching objective since negative datapoints should have near-zero density in the original real-world distribution that the model is trying to mimic. We will refer to models that train using negative data as negative-data generative models, or NDGMs.

Although many existing generative modeling formulations, such as binary class-conditional models, can be simply adapted into NDGMs, specialized NDGMs have also been proposed. The prior state-of-the-art (SOTA) method suffers from two major shortcomings: It never specifically learns the density ratio between the positive and negative data and it often suffers from mode collapse. We conceptualize and test a new NDGM that overcomes these issues through the use of a multi-class discriminative model that learns individual density ratios and a Determinantal-Point-Process (DPP)-based loss that encourages diverse sample sets. Through extensive benchmarking, we demonstrate that our new formulation outperforms both simple baselines and the current state-of-the-art NDGMs on highly non-convex test problems, a variety of real-world engineering tasks, and high-dimensional image-based tasks. Our key contributions are summarized below.

(i) We introduce a new NDGM which significantly outperforms the constraint satisfaction of vanilla generative models while addressing mode collapse issues in current NDGMs. These advancements are enabled by estimating individual density ratios and introducing a diversity-based training loss.

(ii) We evaluate our model on an expansive set of benchmarks including specially-constructed test problems, authentic engineering tasks featuring real-world constraints from engineering standards, and a final high-dimensional topology optimization study. We compare our model to 10 baseline training formulations spanning adversarial models, variational autoencoders, and diffusion models.

(iii) We demonstrate that our model frequently achieves an unmatched tradeoff between constraint satisfaction and distribution learning, in some cases attaining 95-98% lower constraint violation than classic generative models, while achieving top-three distributional similarity scores.

(iv) We show that our NDGM model can significantly outperform vanilla models, generating 1/6 as many constraint-violating samples using only 1/8 as much data. This makes our model an excellent choice in data-constrained problems involving constraints.

## 2 Background

In this section, we introduce key considerations in constraint satisfaction for engineering design, discuss divergence minimization in generative models, then introduce negative-data generative models. For more background and related work, see Appendix C.

### 2.1 Constraints in Engineering and Design

Constraints are ubiquitous in design. A designer creating ship hulls, for example, must adhere to a medley of geometric constraints, performance requirements, and safety regulations from authoritative bodies. Generating constraint-satisfying designs can be exceedingly difficult. As many practitioners turn to data-driven generative models to tackle engineering problems (Regenwetter et al., 2022a), this difficulty remains (Woldseth et al., 2022; Regenwetter et al., 2023). For example, even a generative model that sees 30K examples of valid ship hulls can only generate valid hulls with a 6% success rate in our experiments.

The overwhelming majority of deep generative models in design do not actively consider constraints (Woldseth et al., 2022; Regenwetter et al., 2022a), despite constraint satisfaction being an explicit goal in many of the design problems they address (Oh et al., 2019; Nie et al., 2021; Bilodeau et al., 2022; Chen et al., 2022; Chen & Fuge, 2019; Cheng et al., 2021). Several engineering design datasets feature constraint-violating designs (Regenwetter et al., 2022b; Bagazinski & Ahmed, 2023; Giannone & Ahmed, 2023; Mazé & Ahmed, 2023), and many others have checks for validity (Whalen et al., 2021; Wollstadt et al., 2022), allowing datasets of constraint-violating (negative) designs to be curated. In some cases, datasets of positive examples are even created through search by rejecting and discarding negative samples (Bagazinski & Ahmed, 2023; Regenwetter et al., 2022b), making negative data essentially free. In any problem where negative data is available or can be generated, NDGMs can be applied.

### 2.2 Divergence Minimization in Generative Models

Before discussing divergence minimization in NDGMs, we first discuss divergence minimization in conventional generative models. Let $p_p(x)$ be the (positive) data distribution and $p_\theta(x)$ the distribution sampled by the generative model. Given $N$ samples from $p_p(x)$, the objective of generative modeling is to find a setting $\theta^*$ of $\theta$, such that, for an appropriate choice of discrepancy measure, $p_\theta^* \approx p_p$. A common choice for this discrepancy measure is the Kullback–Leibler or KL divergence:

$$\mathbb{KL}[p_\theta \| p_p] = \int p_\theta(\mathbf{x}) \left[ \log \frac{p_\theta(\mathbf{x})}{p_p(\mathbf{x})} \right] d\mathbf{x}. \tag{1}$$

To minimize the discrepancy, we find $\theta^*$ as the solution to the following optimization problem:

$$\theta^* = \arg\min_\theta \mathbb{KL}[p_\theta \| p_p]. \tag{2}$$

In practice, direct optimization of Eq. (2) is often intractable. As such, it is common in deep generative modeling to learn $\theta$ by using either a tractable lower bound to a slightly different variant of Eq. (2) (Kingma & Welling, 2013; Burda et al., 2015; Ho et al., 2020; Sønderby et al., 2016) or by using plug-in or direct estimators of the divergence measure (Casella & Berger, 2002; Sugiyama et al., 2012a; Gutmann & Hyvärinen, 2010; Srivastava et al., 2017; Goodfellow et al., 2014; Srivastava et al., 2023; Poole et al., 2019). In both of these cases, under certain conditions, as $N \to \infty$, theoretically, it holds that, $\theta \to \theta^*$. However, since $N$ is limited, there remains a finite discrepancy between the model and data distributions. This mismatch often manifests in $p_\theta$ allocating high probability mass in regions where $p_p$ may not have significant empirical support. In domains such as engineering design, where invalid (negative) designs tend to be very close to the valid (positive) designs, this leads to the generation of invalid designs with high probability. This lack of precision underpins the relatively limited success of deep generative models in the engineering design domain (Regenwetter et al., 2023).

**Divergence minimization in GANs.** Generative Adversarial Networks (GANs) (Goodfellow et al., 2014; Arjovsky et al., 2017; Mohamed & Lakshminarayanan, 2016; Srivastava et al., 2017; Nowozin et al., 2016) are a powerful framework for generating realistic and diverse data samples. GANs have two main components: a generator $f_\theta$, which generates samples according to the density $p_\theta$, and a discriminator $f_\phi$, which is a

binary classifier. The generator learns to generate synthetic data samples by transforming random noise into meaningful outputs, while the discriminator aims to distinguish between real and generated samples. The standard GAN loss can be written as:

$$\mathcal{L}(\theta, \phi) = \mathbb{E}_{p_p(\mathbf{x})}[\log f_\phi(\mathbf{x})] + \mathbb{E}_{p_\theta(\mathbf{x})}[1 - \log(f_\phi(\mathbf{x}_\theta))]. \tag{3}$$

Training a GAN involves iterating over $\min_\theta \max_\phi \mathcal{L}(\theta, \phi)$. GANs can also be interpreted in terms of estimating the density ratio (Gutmann & Hyvärinen, 2010; Srivastava et al., 2017) between the data and the generated distribution $r(\mathbf{x}) = p_p(\mathbf{x})/p_\theta(\mathbf{x})$. This ratio can be estimated by a discriminative model as $r_\phi = f_\phi(\mathbf{x})/(1 - f_\phi(\mathbf{x}))$ and $r_\phi = 1$ gives us $p_\theta = p_p$. The optimal discriminator prediction and generator distribution are:

$$f_\phi(\mathbf{x}) = \frac{p_p(\mathbf{x})}{(p_\theta(\mathbf{x}) + p_p(\mathbf{x}))}, \ p_\theta^*(\mathbf{x}) = p_p(\mathbf{x}). \tag{4}$$

**Divergence minimization in other generative models.** Many other types of generative models similarly minimize divergence between $p_\theta$ and $p_p$. These models include popular likelihood-based models like Variational Autoencoders (VAEs) (Kingma & Welling, 2013) and Denoising Diffusion Probabilistic Models (DDPMs) (Ho et al., 2020). We will not discuss the mathematics behind divergence minimization for these likelihood-based models, but we do benchmark several variants in our results. In general, we refer to unaugmented GANs, VAEs, and DDPMs as 'vanilla' models throughout the paper.

## 2.3 Negative-Data Generative Models (NDGMs)

In this section, we discuss the NDGM framework. We explain how generative models can be adjusted to exploit negative data to improve constraint satisfaction. Let $p_n$ denote the negative distribution i.e., the distribution of constraint-violating datapoints. A key assumption in the negative data formulation is that $p_p$ and $p_n$ have nearly mutually exclusive support. Instead of training using only the positive distribution $p_p$, we now seek to train a generative model using both $p_p$ and $p_n$. In this section, we discuss several existing methods to do so. These methods range from simple baselines like auxiliary classifiers to state-of-the-art formulations like discriminator overloading.

### 2.3.1 Class Conditioning

Class conditional modeling is a simple approach to incorporate constraints into generative models, which is popular in many design generation problems (Nie et al., 2021; Behzadi & Ilieş, 2021; Mazé & Ahmed, 2023; Shin et al., 2023; Heyrani Nobari et al., 2021). In conditional modeling, a generative model typically conditions on the constraints denoted as $\mathbf{c}$ and learns a conditional distribution, $p(\mathbf{x}|\mathbf{c})$, where $\mathbf{x}$ represents the generated output. 'Off-the-shelf' class-conditional models can be simple NDGMs, where the positive and negative data each constitute one class. During inference, the model attempts to satisfy constraints by generating conditionally positive samples. Broadly speaking, the negative data formulation for generative models can be seen as a specific case of class-conditional generation. However, as we demonstrated in Appendix A, generic class-conditional training formulations for generative models are not as effective as specialized NDGMs.

### 2.3.2 Classifier-Augmented Generative Models

Another common approach to actively satisfy constraints using generative models involves training a supervised model to predict constraint satisfaction. After training, this supervised model is subsequently queried during generative model training or inference. Often, this model predicts constraint violation likelihood, though it can also predict intermediates that are combined in a more complex constraint check (Wang et al., 2022). Typically, this classifier $f_\psi$ learns:

$$f_\psi(\mathbf{x}) = \frac{p_n(\mathbf{x})}{p_n(\mathbf{x}) + p_p(\mathbf{x})}. \tag{5}$$

This frozen classifier can be incorporated into the training of a generative model by adding an auxiliary loss, $\mathcal{L}_{PC}$ to the generative model's loss, $\mathcal{L}_{GM}$ to calculate a total loss, $\mathcal{L}_{Tot} = \mathcal{L}_{GM} + \lambda \mathcal{L}_{PC}$, as in (Regenwetter & Ahmed, 2022). Here, $\lambda$ is some weighting parameter and $\mathcal{L}_{PC}$ is expressed as:

$$\mathcal{L}_{PC} = \mathbb{E}_{p_\theta(\mathbf{x})}[\log f_\psi(\mathbf{x})]. \tag{6}$$

Pre-trained classifiers for constraints can also be applied during inference in certain models, such as in diffusion model guidance (Mazé & Ahmed, 2023; Giannone & Ahmed, 2023). They can also work alongside a trained generative model as a rejection-sampling postprocessing step, which is discussed in Appendix D.

### 2.3.3 Discriminator Overloading (DO)

Discriminator overloading is a technique to directly incorporate negative data into GAN model training. This formulation was proposed in two of the first papers to train a generative model using both positive and negative data (though we have made slight modifications for generality): Rumi-GAN (Asokan & Seelamantula, 2020) and Negative Data Augmentation GAN (NDA-GAN) (Sinha et al., 2021). We refer to these formulations as 'discriminator overloading' since the discriminator is 'overloaded' by learning to discriminate between (1) positives and (2) fakes or negatives. As such, the discriminator estimates:

$$f_\phi(\mathbf{x}) = \frac{p_p(\mathbf{x})}{\lambda p_\theta(\mathbf{x}) + (1 - \lambda)p_n(\mathbf{x}) + p_p(\mathbf{x})}, \tag{7}$$

with $\lambda$ being a weighting parameter. As usual, the generator attempts to generate samples that are classified as real, in this case indicating that they look similar to positive data and dissimilar to negative data. The loss is expressed as:

$$\mathcal{L}(\theta, \phi) = \mathbb{E}_{p_p(\mathbf{x})}[\log f_\phi(\mathbf{x})] + \mathbb{E}_{p_\theta(\mathbf{x})}[1 - \log(f_\phi(\mathbf{x}_\theta))] + \mathbb{E}_{p_n(\mathbf{x})}[1 - \log(f_\phi(\mathbf{x}))]. \tag{8}$$

In our benchmarks, we refer to GAN models trained using discriminator overloading as GAN-DO models. Discriminator overloading is effective and can be considered the existing state-of-the-art in NDGMs. However, as we will show, discriminator overloading has significant shortcomings which we address in our new training formulation.

## 3 Methodology

NDGMs suffer from a widespread tendency toward mode collapse seen across a variety of baselines. Critically, the state-of-the-art GAN-DO training formulation is particularly prone to this issue due to its conflation of fakes and negatives during training. To address these shortcomings, we propose a new NDGM training formulation that introduces two new innovations: First, and central to our approach, we propose to learn the ratios between the positive, negative, and fake distributions individually, rather than conflating the negatives and fakes. Second, we add an auxiliary diversity-based training objective to NDGM training which directly mitigates mode collapse. We describe these innovations in detail below.

### 3.1 Learning Individual Density Ratios Using a Multi-Class Discriminator

GAN-DO learns a density ratio between $p_p$ and an amalgamation of $p_n$ and $p_\theta$. Rather than conflating the negative data and fake samples, we instead advocate to learn pairwise density ratios between the three distributions. Noting that multi-class classifiers are strong density ratio estimators (Srivastava et al., 2023), we propose to learn these ratios using a multi-class discriminator. This multi-class discriminator model learns to discriminate three classes: positive, negative, and fake, thereby learning their pairwise density ratios:

$$f_{\phi,c}(\mathbf{x}) = \frac{p_c(\mathbf{x})}{p_p(\mathbf{x}) + p_\theta(\mathbf{x}) + p_n(\mathbf{x})} \ \forall \ c \in p, n, \theta. \tag{9}$$

Note that $f_{\phi,p}$ is a reweighted version of Eq. 7. Though this multi-class formulation is similar to discriminator overloading, instead of showing the discriminator a weighted amalgamation of fakes and negatives (as in DO), the multi-class discriminator instead treats fakes and negatives as separate classes, and can potentially refine its knowledge by distinguishing them. Complemented by a generator model which tries to maximize $f_{\phi,p}(x_\theta)$, this classifier fulfills the role of the discriminator in an adversarial training formulation. Notably, $f_{\phi,p}/f_{\phi,n}$ estimates $p_p/p_n$, which is never directly learned in the discriminator overloading formulation.

We also note that there are numerous other solutions to learn density ratios between positive and negative data distributions. For example, a direct estimator of $p_p/p_n$ can operate alongside the classic discriminator in a two-discriminator NDGM variant. We discuss the general motivation for density ratio learning in NDGMs in Appendix C and the mathematical formulation behind the double discriminator variant in Appendix E.

### 3.2 Addressing Mode Collapse Using a Diversity-Based Loss

When augmented with validity-based training objectives, NDGMs tend to collapse in valid regions of the sample space. This effectively allows them to excel in validity and precision, but struggle with recall and diversity. This tendency arises because a conservative NDGM will avoid regions of the distribution near the constraint boundary, resulting in incomplete coverage. One approach to improve recall is to explicitly encourage diversity of generated samples. Diversity is often a desired goal in generative modeling for engineering design applications (Regenwetter et al., 2023). As Chen & Ahmed (2021a) note, incorporating diversity can also help models generalize and avoid mode collapse. Determinantal Point Process (DPP)-based diversity measures (Kulesza et al., 2012) have been used in a variety of generative applications in design (Chen & Ahmed, 2021b; Nobari et al., 2021; Regenwetter & Ahmed, 2022) and elsewhere (Elfeki et al., 2019; Mothilal et al., 2020).

The DPP loss is calculated using a positive semi-definite DPP kernel $S$. Entries of this matrix are calculated using some modality- and problem-dependent similarity kernel, such as the Euclidean distance kernel. The $(i, j)^{th}$ element of S can be expressed in terms of the similarity kernel $k$ and samples $x_i$ and $x_j$ as $S_{i,j} = k(x_i, x_j)$, and the loss as:

$$\mathcal{L}_{Div}(\mathbf{x}) = -\frac{1}{B} \operatorname{logdet}(S(\mathbf{x})) = -\frac{1}{B} \sum_{i=1}^{B} \log \lambda_i, \tag{10}$$

where $\lambda_i$ is the i-th eigenvalue of $L$ and $B$ is the number of samples in the batch. The loss is incorporated by appending it to the overall loss term of the generative model $\mathcal{L}_{GM}$:

$$\mathcal{L}_{Tot}(\mathbf{x}) = \mathcal{L}_{GM}(\mathbf{x}) + \gamma \, \mathcal{L}_{Div}(\mathbf{x}), \tag{11}$$

where $\gamma$ is a weighting parameter modulating the diversity loss contribution. Adding this loss to NDGMs can help the generative model achieve better coverage, an observation demonstrated in our experiments below.

### 3.3 GAN with Multi-Class Discriminator and Diversity Loss

We combine the above innovations into a new training formulation called GAN-MDD (Multiclass Discriminator + Diversity). The generator trains to generate samples which minimize:

$$\mathcal{L}_\theta(\mathbf{x}_\theta, \phi) = f_{\phi,p}(\mathbf{x}_\theta) + \gamma \, \mathcal{L}_{Div}(\mathbf{x}_\theta). \tag{12}$$

To illustrate the effectiveness of this specific formulation, we include ablation studies on diversity-augmented generation in Section 4.3. Training pseudocode is included in Appendix B. In the upcoming Section 4, we extensively benchmark GAN-MDD against baseline models and GAN-DO on a variety of problems.

## 4 Experiments and Results

We now present experiments on (i) 2D densities, where we benchmark 11 different models including ours, the SOTA, baseline NDGMs, and vanilla models; (ii) A dozen diverse engineering tasks featuring real-world constraints from regulatory authorities and engineering standards, where we demonstrate the potency of our approach (iii) a high-dimensional free-form topology optimization problem where we explore the impact of negative data quality. We also perform ablation studies and data-efficiency studies. We include significant additional visualization and statistical testing for experimental results in Appendix A. Details on datasets and model training are included in Appendix F.

### 4.1 Extensive Benchmarking on 2D Densities with Constraints

We first construct a pair of 2D densities as easy-to-visualize tests for NDGM models to visually showcase their characteristics. Despite being low-dimensional and relatively structured, these problems are very challenging for vanilla models and NDGMs alike. Notably, constraint-violating regions appear in close proximity to high-density regions of the valid data distribution. For a model to succeed, precise estimation of constraint boundaries is essential.

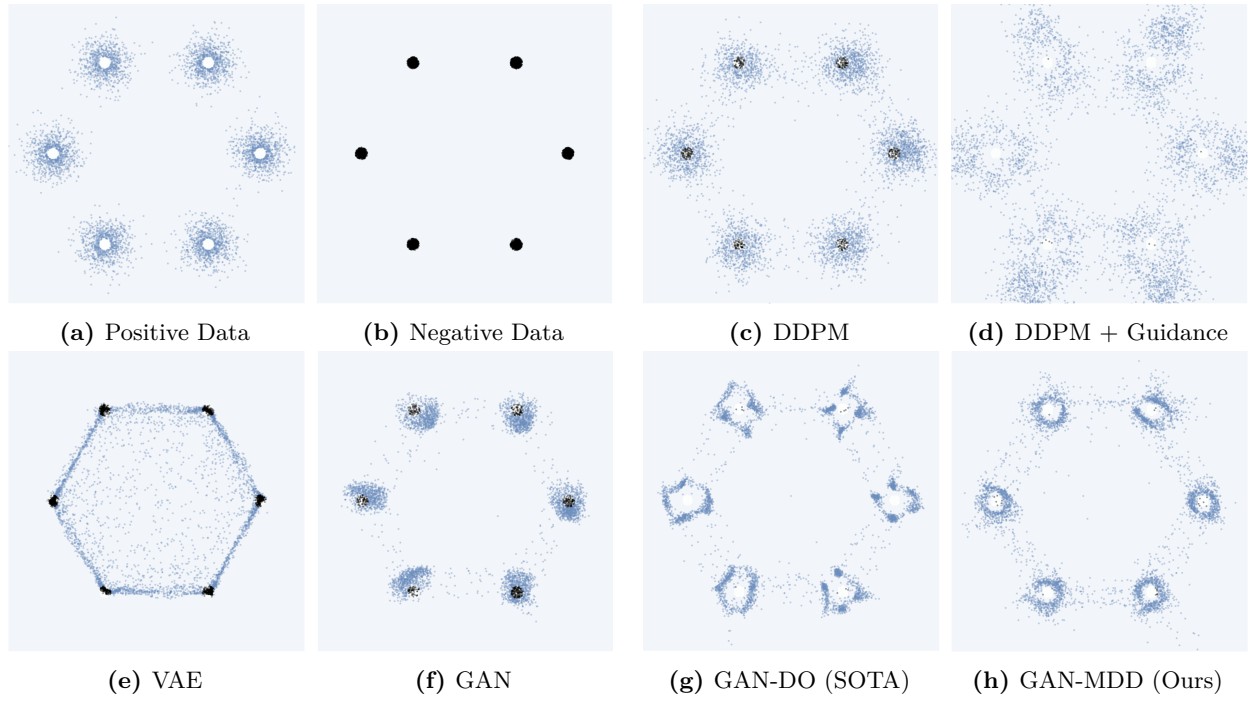

**Figure 2:** Generated distributions from select generative models on `Problem 1`, a mixture of Gaussians with invalid region in the center of each mode. Positive data points and samples are shown in blue and negative ones in black. Our proposed NDGM model, GAN-MDD learns the distribution most faithfully.

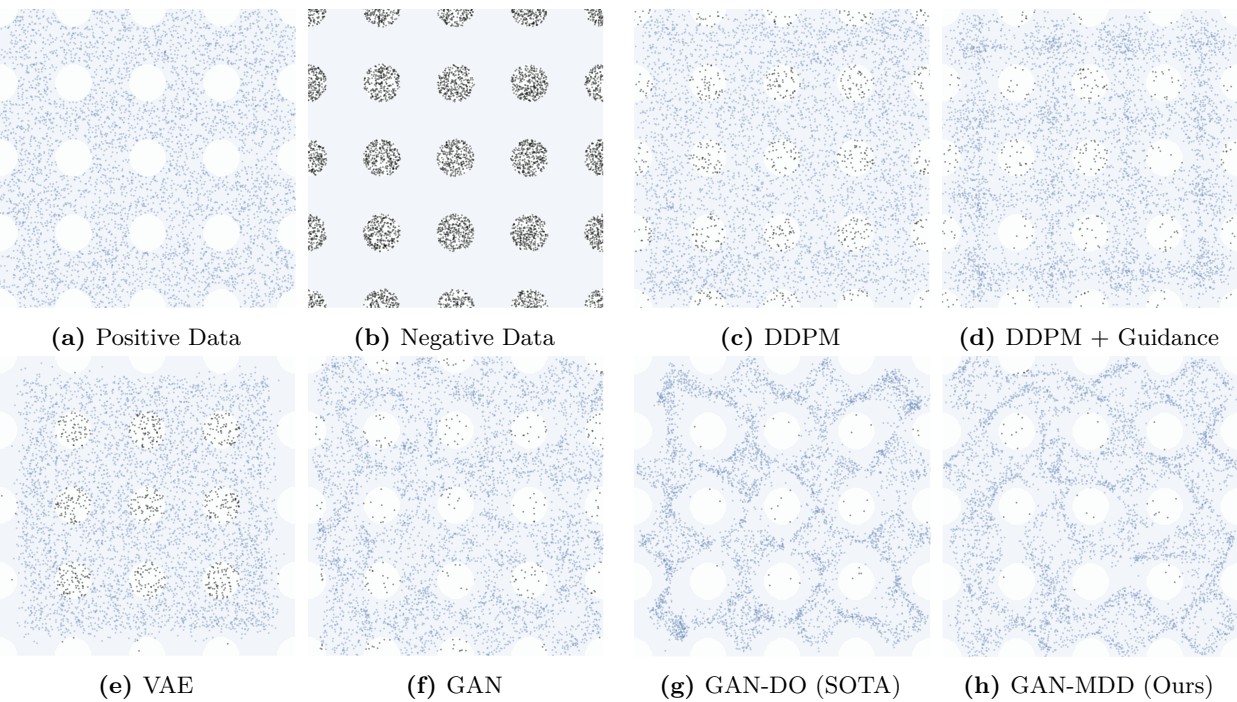

**Figure 3:** Generated distributions from select generative models on `Problem 2`, a uniform distribution with many circular invalid regions. Positive data points and samples are shown in blue and negative ones in black. Our proposed NDGM model, GAN-MDD learns the distribution most faithfully.

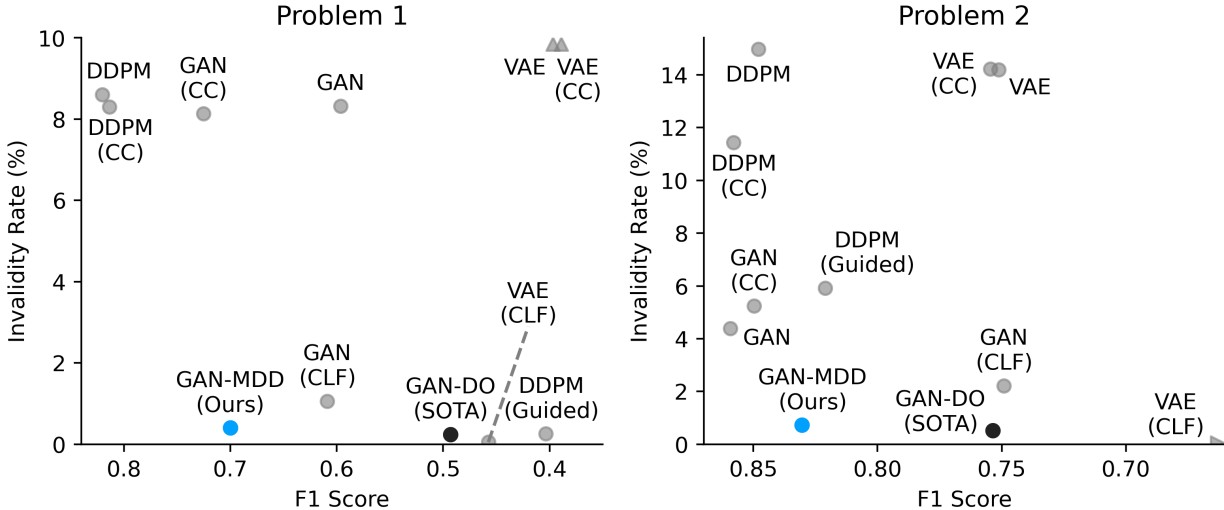

**Figure 4:** Comparison of F1 scores (↑) and invalidity rates (↓) for benchmarked models on `Problem 1` (left) and `Problem 2` (right). Mean scores over six instantiations are plotted. Scores closer to the bottom left are more optimal. Triangular markers indicate that the score lies off the plot in the indicated direction. Class conditioning, classifier loss, and guidance are denoted with (CC), (CLF), and (Guided), respectively.

**Models.** We test 11 variants of GAN, VAE, and DDPM models. Among these are: three vanilla models (GAN, VAE, DDPM), trained only on positive data; three class conditional models (GAN, VAE, DDPM), trained on both positive and negative data in a binary class conditional setting as in Section 2.3.1; three models augmented with a frozen pre-trained validity classifier to steer models during training (GAN, VAE) or during inference using guidance (DDPM), as in Section 2.3.2; GAN with discriminator overloading as in NDA-GANs (Sinha et al., 2021) and Rumi-GANs (Asokan & Seelamantula, 2020) (GAN-DO) (Section 2.3.3); Our multi-class-discriminator GAN with diversity (GAN-MDD). Each model is tested six times.

**Metrics.** We seek to measure each model's reliability in constraint-satisfaction, distribution learning ability, and ability to avoid mode collapse. We therefore score each model on three metrics: 1) Invalidity – the fraction of generated samples that violate the constraints (negative samples). 2) $F_1$ score for generative models, a common distributional similarity metric proposed in Sajjadi et al. (2018). 3) DPP Diversity score, a metric which highlights mode collapse and is measured as described in Section 3.2. The ideal model maximizes F1 and minimizes invalidity and DPP diversity.

**Results.** Figures 2 and 3 plot the datasets and the generated distributions of a subset of the tested models. All models' distributions are plotted in Figures 13 and 14 in Appendix A. Compared to baseline models, our GAN-MDD model learns the best estimate of the data distribution while avoiding constraint violation.

Figure 4 plots mean F1 scores and invalidity rates on both problems for all models. The scores confirm that our GAN-MDD achieves an optimal tradeoff between statistical similarity and constraint satisfaction. Table 7 contains the numerical scores, with mean and standard deviations over the six training runs, as well as statistical significance testing using 2-sample t-tests. Our GAN-MDD model significantly outperforms the state-of-the-art GAN-DO in distributional similarity and diversity while achieving similar constraint-satisfaction performance.

## 4.2 Is Negative Data More Valuable than Positive Data?

Given an infinite amount of data, model capacity, and computational resources, generative models can theoretically approach an exact recovery of the underlying data distribution, $p_p$. In practical scenarios, however, data and computational throughput are limited, particularly in fields like engineering design and scientific research. Thus, simply increasing the volume of data is not a viable strategy to improve constraint satisfaction. Fortunately, we find that NDGMs can be significantly more data-efficient than vanilla generative models, giving them a significant advantage in data-constrained domains.

**Table 1:** Study of **invalidity rates** for GAN-MDD trained with different numbers of positive datapoints ($N_p$) and negative datapoints ($N_n$). Note that GAN-MDD without negative data ($N_n = 0$) trains as a vanilla GAN. Diversity loss is turned off. Scores are averaged over four instantiations. **Lower is better.** NDGMs can generate significantly fewer constraint-violating samples, even when trained on orders of magnitude less data.

**Figure 5:** Comparison of invalidity rate (↓) and number of datapoints (↓) under various mixtures of positive and negative data on `Problem 2`. Points are labeled using their proportion of negative data. Any amount of negative data tested improves performance. Interestingly, a small proportion of negative data (20%) yields better constraint satisfaction than higher proportions in this problem.

**Problem 1**

|            | $N_p = 1K$ | $N_p = 4K$ | $N_p = 16K$ |
|------------|-----------|-----------|------------|
| $N_n = 0$  | 10.7%     | 11.6%     | 11.9%      |
| $N_n = 1K$ | 2.0%      | 0.7%      | 1.5%       |
| $N_n = 4K$ | 0.5%      | 0.5%      | 0.5%       |
| $N_n = 16K$| 0.5%      | 0.7%      | 0.6%       |

**Problem 2**

|            | $N_p = 1K$ | $N_p = 4K$ | $N_p = 16K$ |
|------------|-----------|-----------|------------|
| $N_n = 0$  | 6.0%      | 3.2%      | 3.3%       |
| $N_n = 1K$ | 2.3%      | 1.6%      | 1.9%       |
| $N_n = 4K$ | 0.6%      | 0.7%      | 0.7%       |
| $N_n = 16K$| 0.5%      | 0.2%      | 0.2%       |

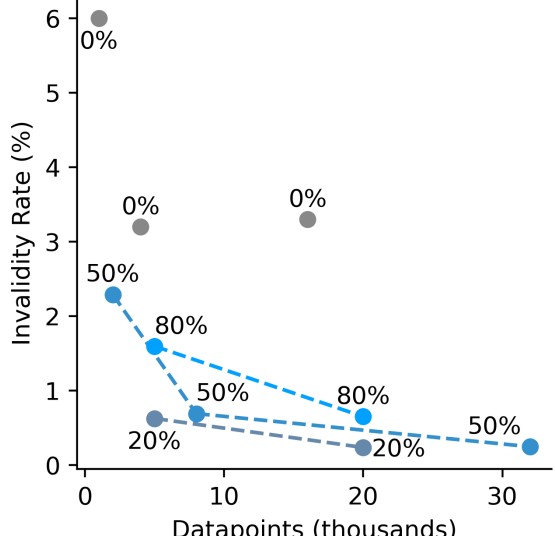

In Table 1 and Figure 5, we explore how the number of positive and negative training datapoints ($N_p$ and $N_n$, respectively) affect invalidity rates on the 2D test problems. Full scores with standard deviations and statistical tests are included in Table 8. To remove confounding factors, we benchmark a fixed-size GAN-MDD model without diversity loss (note: GAN-MDD is equivalent to a vanilla GAN when trained with no negative data). We test four model instantiations per dataset mixture.

We find that for pure positive datasets, adding more data yields diminishing returns, while mixing in negative data drastically improves constraint satisfaction. For example, in `Problem 1`, with $N_p = 1K$, $N_n = 1K$, GAN-MDD generates 1/6 as many invalid samples with 1/8 as much data compared to $N_p = 16K$, $N_n = 0$. On `Problem 2` with $N_p = 4K$, $N_n = 1K$, GAN-MDD generates 1/5 as many invalid samples with 1/3 as much data compared to $N_p = 16K$, $N_n = 0$. Since NDGMs can be significantly more data-efficient than vanilla models, practitioners seeking to improve their generative models may attain much more value by collecting even a small amount of negative data, rather than additional positive data. This study also prompts an interesting research question: What is the optimal ratio of positive to negative data? A rigorous answer, though certainly problem dependent, could lead to more efficient dataset generation and curation.

### 4.3 Ablation Study: Examining the Effect of Diversity Loss

In many negative data settings, a diversity-based training objective can serve to modulate the tradeoff between constraint satisfaction and distributional similarity. To showcase this, we examine the performance of GAN-MDD over a sweep of diversity loss weights ($\gamma$ from Eq. 11). Simultaneously, we examine the performance of GAN-DO augmented with our DPP-based diversity loss. These experiments illustrate the effect of the diversity loss in modulating the tradeoff between constraint satisfaction and distributional similarity. They also serve as an experimental ablation study to confirm the effect of our two NDGM innovations: the multi-class discriminator (versus discriminator overloading) and the diversity-based loss.

We visually showcase distributions generated by GAN-MDD and GAN-DO under different weights of $\gamma$ in Figures 6 and 7. As expected, higher diversity yields better distributional similarity but poorer constraint

satisfaction in both models across both problems. GAN-MDD generates very neat distributions with higher diversity, creating the most visually similar sampled distribution compared to the ground-truth distribution. In contrast, GAN-DO never truly captures the ground truth distribution, despite overcoming its more egregious distribution collapse issues.

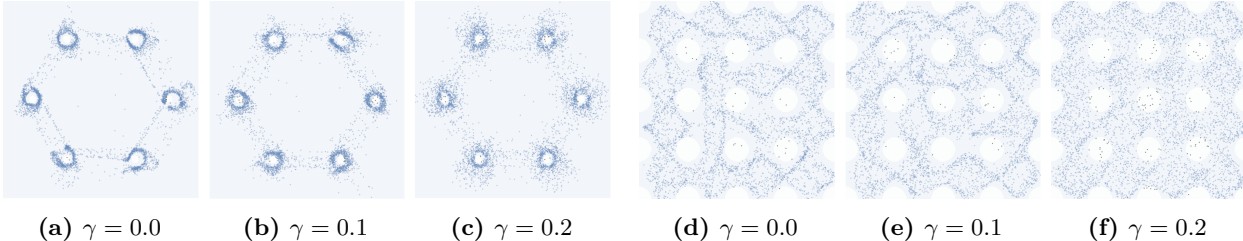

**(a)** $\gamma = 0.0$  **(b)** $\gamma = 0.1$  **(c)** $\gamma = 0.2$  **(d)** $\gamma = 0.0$  **(e)** $\gamma = 0.1$  **(f)** $\gamma = 0.2$

**Figure 6:** Generated distributions from by our **GAN-MDD** for `Problem 1` (a-c) and `Problem 2` (d-f), demonstrating effect of diversity loss weight, $\lambda$. Diversity weight elegantly modulates the tradeoff between distributional similarity and constraint satisfaction.

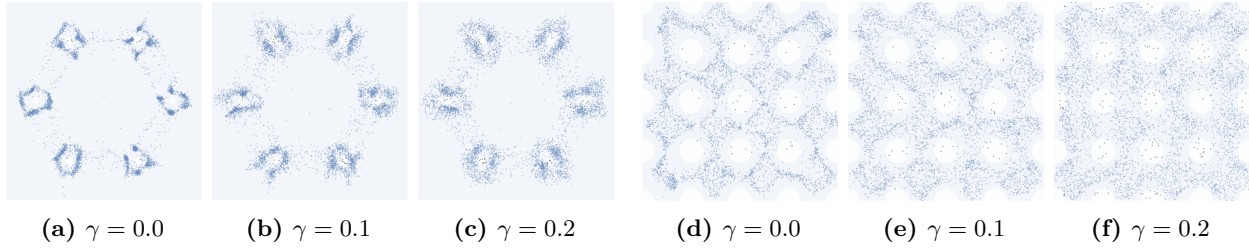

**(a)** $\gamma = 0.0$  **(b)** $\gamma = 0.1$  **(c)** $\gamma = 0.2$  **(d)** $\gamma = 0.0$  **(e)** $\gamma = 0.1$  **(f)** $\gamma = 0.2$

**Figure 7:** Generated distributions from the baseline **GAN-DO** for `Problem 1` (a-c) and `Problem 2` (d-f), demonstrating effect of diversity loss weight, $\lambda$. Although adding diversity loss eases mode collapse issues, GAN-DO with large diversity weight struggles with high invalidity.

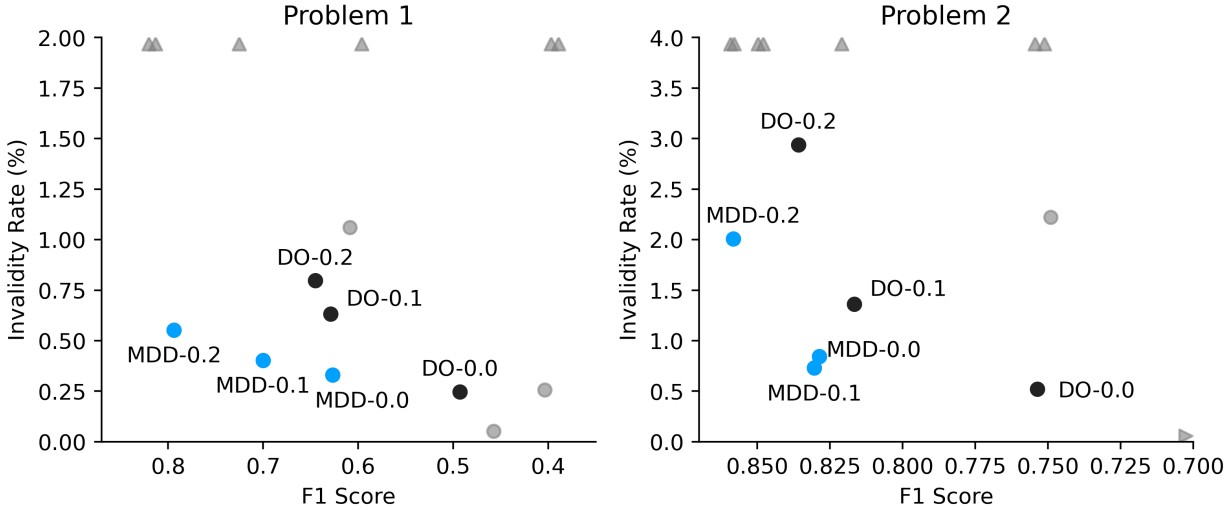

**Figure 8:** Comparison of GAN-MDD and GAN-DO invalidity rates ($\downarrow$) and F1 scores ($\uparrow$) for a variety of diversity weights on `Problem 1` (left) and `Problem 2` (right). Gray markers indicate scores for other benchmarked models. Mean scores over six instantiations are plotted. Scores closer to the bottom left are more optimal. Triangular markers indicate that the score lies off the plot in the indicated direction. Exact scores are included in Table 9.

We present a summary of numerical scores in Figure 8, with full scores in Table 9. These scores illustrate GAN-MDD's widespread dominance across a variety of diversity weights. Across all weights tested, GAN-MDD achieves significantly better distributional similarity scores than GAN-DO. Furthermore, for every nonzero diversity weight, GAN-MDD achieves significantly better constraint satisfaction.

## 4.4 Negative-Data Generative Models Excel in Engineering Tasks

Generative models are commonly used to tackle engineering problems with constraints (Oh et al., 2019; Nie et al., 2021), but are often criticized for their inability to satisfy them (Woldseth et al., 2022; Regenwetter et al., 2023). To assess how NDGMs fare in real engineering problems, we have curated a benchmark of a dozen diverse engineering tasks, which are discussed in detail in Appendix F. These problems span numerous engineering disciplines including assorted industrial design tasks (compression spring, gearbox, heat exchanger, pressure vessel), structural and material design tasks (Ashby chart, cantilever beam, reinforced concrete, truss, welded beam), and several complex high-level design problems: Ship hulls with hydrodynamic constraints; bike frames with loading requirements; automobile chassis with performance requirements in impact testing. A variety of constraints are applied, including engineering standards from authoritative bodies like the American Concrete Institute (ACI), the American Society of Mechanical Engineers (ASME), and the European Enhanced Vehicle-Safety Committee (EEVC). As a select example, we visualize several positive and negative datapoints from the FRAMED bike frame dataset (Regenwetter et al., 2022b) in Figure 9.

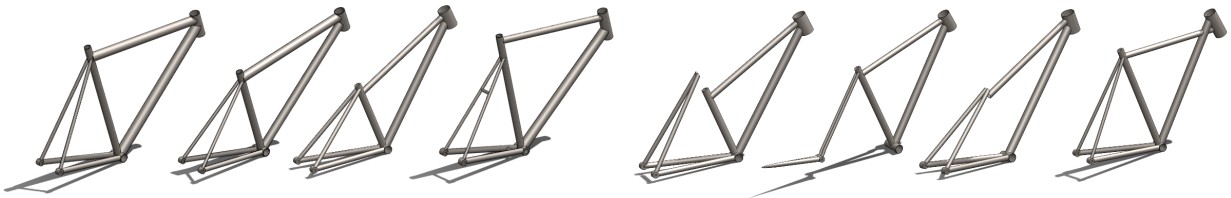

**(a)** Constraint-satisfying (positive) bike frames      **(b)** Constraint-violating (negative) bike frames

**Figure 9:** Example visualization for the bike frame engineering problem. The FRAMED bike frame dataset describes a 37-dimensional 3D parametric CAD problem that defines dozens of geometric constraints (disconnected components, negative tube thickness, etc.). Vanilla generative models generate a variety of constraint-violating bikes, while NDGMs like GAN-MDD reliably generate valid bikes.

We evaluate a vanilla GAN, the state-of-the-art GAN-DO, and our GAN-MDD. We measure invalidity rate, F1 score, and DPP diversity scores over seven training runs for each problem. For each problem and metric, we evaluate whether models are significantly ($p < 0.05$) superior to one another using one-sided 2-sample t-tests. A summary of benchmarking results is presented in Table 2, with detailed results in Tables 3, 4, and 5. We summarize key results as follows:

- **Invalidity Score**: GAN-DO and GAN-MDD are the highest performers in invalidity score, significantly outperforming a vanilla GAN in almost all problems. GAN-DO is more often the top performer, but sometimes falls short of GAN-MDD.

- **F1 Score**: GAN-MDD is the winner in distributional similarity by a small margin over the GAN. However, the state-of-the-art GAN-DO falls significantly short of both the GAN and GAN-MDD in many problems.

- **Diversity Score**: GAN-MDD is the overwhelming winner in diversity score, significantly outperforming both the GAN and GAN-DO in the majority of problems, and achieving the highest mean score in all but one problem.

In summary, our GAN-MDD is able to achieve significantly better sample validity and diversity than vanilla models while maintaining generally comparable distributional similarity scores. In contrast, the state-of-the-art GAN-DO is able to achieve marginally higher sample validity, but does so at the expense of distributional similarity and sample diversity. Unless constraint satisfaction is favored above all else, we foresee that GAN-MDD will offer a more optimal tradeoff of performance considerations for most users in most problems.

**Table 2:** Invalidity rates, F1 scores, and diversity scores for 12 engineering datasets. We benchmark a vanilla GAN, a GAN with discriminator overloading (DO), and a GAN with a multi-class discriminator and diversity loss (MDD). Mean scores over seven training instances are presented, with best scores for each metric on each problem underlined. Lower invalidity rates and diversity scores are better, while higher F1 scores are better. For each dataset and metric, we compare models in a pairwise fashion and identify when a model outperforms a competitor to a statistically significant degree (details in Tables 3-5). These 'pairwise wins' are tallied across all problems and presented in the last row.

| Dataset | Invalidity (%) ( ↓ ) | | | F1 Score ( ↑ ) | | | Diversity Score ( ↓ ) | | |
|---|---|---|---|---|---|---|---|---|---|
| | GAN (base) | DO (sota) | MDD (ours) | GAN (base) | DO (sota) | MDD (ours) | GAN | DO (sota) | MDD (ours) |
| Ashby Chart | 1.54 | 1.12 | 0.63 | 0.959 | 0.960 | 0.922 | 10.30 | 10.26 | 10.19 |
| Bike Frame | 4.77 | 2.85 | 5.46 | 0.681 | 0.684 | 0.731 | 3.17 | 3.06 | 1.28 |
| Cantilever Beam | 4.16 | 2.51 | 3.00 | 0.845 | 0.818 | 0.875 | 1.25 | 1.94 | 0.94 |
| Car Impact | 4.78 | 1.92 | 3.84 | 0.883 | 0.844 | 0.893 | 0.54 | 1.12 | 0.36 |
| Comp. Spring | 1.49 | 0.77 | 1.06 | 0.960 | 0.956 | 0.962 | 10.84 | 10.83 | 10.84 |
| Concrete Beam | 1.03 | 0.16 | 1.14 | 0.957 | 0.954 | 0.956 | 10.33 | 10.31 | 10.29 |
| Gearbox | 0.33 | 0.02 | 0.07 | 0.899 | 0.872 | 0.891 | 5.32 | 5.89 | 4.56 |
| Heat Exchanger | 5.35 | 3.77 | 3.68 | 0.876 | 0.869 | 0.867 | 4.82 | 5.12 | 4.64 |
| Pressure Vessel | 1.30 | 0.11 | 0.95 | 0.947 | 0.944 | 0.932 | 9.42 | 9.39 | 9.35 |
| Ship Hull | 93.97 | 93.54 | 92.05 | 0.769 | 0.708 | 0.713 | 11.06 | 11.09 | 11.05 |
| Three-Bar Truss | 0.32 | 0.00 | 0.34 | 0.938 | 0.948 | 0.957 | 14.32 | 14.28 | 14.19 |
| Welded Beam | 1.74 | 0.67 | 0.53 | 0.955 | 0.936 | 0.850 | 9.44 | 9.37 | 9.30 |
| Pairwise Wins* | 0 | **14** | 9 | 7 | 2 | **9** | 3 | 3 | **17** |

*Number of models statistically significantly outperformed in pairwise comparisons. Details in Tables 3-5.

**Table 3:** Means and standard deviations of **invalidity rate** for engineering datasets over seven tests. Lower scores are better. Problems are sorted by GAN's invalidity rate. A model's symbol (✓/✗/†) is shown if statistically significant in a pairwise comparison. Best mean scores are **bolded**.

| | GAN (baseline) | DO (sota) | MDD (ours) | DO † GAN ✗ | MDD ✓ GAN ✗ | MDD ✓ DO † |
|---|---|---|---|---|---|---|
| Three-Bar Truss | 0.32±0.51% | **0.00±0.00**% | 0.34±0.49% | – | – | † |
| Gearbox | 0.33±0.09% | **0.02±0.02**% | 0.07±0.05% | † | ✓ | † |
| Concrete Beam | 1.03±0.97% | **0.16±0.14**% | 1.14±0.46% | † | – | † |
| Pressure Vessel | 1.30±0.32% | **0.11±0.07**% | 0.95±0.29% | † | ✓ | † |
| Comp. Spring | 1.49±1.00% | **0.77±0.67**% | 1.06±0.58% | – | – | – |
| Ashby Chart | 1.54±0.97% | 1.12±0.39% | **0.63±0.17**% | – | ✓ | ✓ |
| Welded Beam | 1.74±0.89% | 0.67±0.39% | **0.53±0.14**% | † | ✓ | – |
| Cantilever Beam | 4.16±0.87% | **2.51±0.79**% | 3.00±0.62% | † | ✓ | – |
| Bike Frame | 4.77±1.21% | **2.85±0.63**% | 5.46±3.39% | † | – | † |
| Car Impact | 4.78±0.55% | **1.92±0.48**% | 3.84±0.78% | † | ✓ | † |
| Heat Exchanger | 5.35±1.00% | 3.77±0.70% | **3.68±0.82**% | † | ✓ | – |
| Ship Hull | 93.97±0.64% | 93.54±0.97% | **92.05±2.31**% | – | ✓ | – |

**Table 4:** Means and standard deviations of **F1 score** for engineering datasets over seven tests. Higher scores are better. Problems are sorted by GAN's F1 score. A model's symbol (✓/✗/†) is shown if statistically significant in a pairwise comparison. Best mean scores are **bolded**.

| | GAN (baseline) | DO (sota) | MDD (ours) | DO † GAN ✗ | MDD ✓ GAN ✗ | MDD ✓ DO † |
|---|---|---|---|---|---|---|
| Comp. Spring | 0.960±0.003 | 0.956±0.004 | **0.962±0.005** | ✗ | – | ✓ |
| Ashby Chart | 0.959±0.007 | **0.960±0.005** | 0.922±0.014 | – | ✗ | † |
| Concrete Beam | **0.957±0.002** | 0.954±0.004 | 0.956±0.005 | – | – | – |
| Welded Beam | **0.955±0.006** | 0.936±0.013 | 0.850±0.025 | ✗ | ✗ | † |
| Three-Bar Truss | 0.938±0.022 | 0.948±0.012 | **0.957±0.005** | – | ✓ | ✓ |
| Pressure Vessel | **0.947±0.012** | 0.944±0.013 | 0.932±0.013 | – | ✗ | – |
| Gearbox | **0.899±0.023** | 0.872±0.021 | 0.891±0.018 | ✗ | – | ✓ |
| Car Impact | 0.883±0.017 | 0.844±0.041 | **0.893±0.010** | ✗ | – | ✓ |
| Heat Exchanger | **0.876±0.035** | 0.869±0.023 | 0.867±0.021 | – | – | – |
| Cantilever Beam | 0.845±0.038 | 0.818±0.027 | **0.875±0.018** | – | ✓ | ✓ |
| Ship Hull | **0.769±0.082** | 0.708±0.273 | 0.713±0.248 | – | – | – |
| Bike Frame | 0.681±0.030 | 0.684±0.025 | **0.731±0.015** | – | ✓ | ✓ |

**Table 5:** Means and standard deviations of **diversity score** for engineering datasets over seven tests. Lower scores are better. Problems are sorted by GAN's diversity score. A model's symbol (✓/✗/†) is shown if statistically significant in a pairwise comparison. Best mean scores are **bolded**.

| | GAN (baseline) | DO (sota) | MDD (ours) | DO † GAN ✗ | MDD ✓ GAN ✗ | MDD ✓ DO † |
|---|---|---|---|---|---|---|
| Car Impact | 0.54±0.10 | 1.12±0.39 | **0.36±0.06** | ✗ | ✓ | ✓ |
| Cantilever Beam | 1.25±0.38 | 1.94±0.55 | **0.94±0.15** | ✗ | ✓ | ✓ |
| Bike Frame | 3.17±0.40 | 3.06±0.29 | **1.28±0.24** | – | ✓ | ✓ |
| Heat Exchanger | 4.82±0.41 | 5.12±0.21 | **4.64±0.45** | – | – | ✓ |
| Gearbox | 5.32±0.20 | 5.89±0.26 | **4.56±0.35** | ✗ | ✓ | ✓ |
| Pressure Vessel | 9.42±0.02 | 9.39±0.02 | **9.35±0.02** | † | ✓ | ✓ |
| Welded Beam | 9.44±0.03 | 9.37±0.02 | **9.30±0.08** | † | ✓ | ✓ |
| Concrete Beam | 10.33±0.07 | 10.31±0.03 | **10.29±0.05** | – | – | – |
| Ashby Chart | 10.30±0.03 | 10.26±0.02 | **10.19±0.03** | † | ✓ | ✓ |
| Comp. Spring | 10.84±0.07 | **10.83±0.04** | 10.84±0.08 | – | – | – |
| Ship Hull | 11.06±0.78 | 11.09±0.85 | **11.05±0.72** | – | – | – |
| Three-Bar Truss | 14.32±0.07 | 14.28±0.08 | **14.19±0.03** | – | ✓ | ✓ |

### 4.5 Examining Negative Data Quality in High-Dimensional Constrained Engineering Problems

Having tested a variety of tabular engineering problems, we next consider whether our proposed methods can translate to higher-dimensional domains such as images. We examine a common engineering design problem known as topology optimization (TO), which seeks to optimally distribute material in a spatial domain to achieve a certain objective (often minimizing mechanical compliance) (Sigmund & Maute, 2013). Simply put, TO is often used to create structures with high rigidity and low weight. The use of generative models for TO is very popular (Shin et al., 2023), but existing methods have been criticized for significant shortcomings (Woldseth et al., 2022) related to constraint satisfaction, such as generated topologies not being fully connected. Disconnected topologies tend to be highly sub-optimal and are impractical to fabricate.

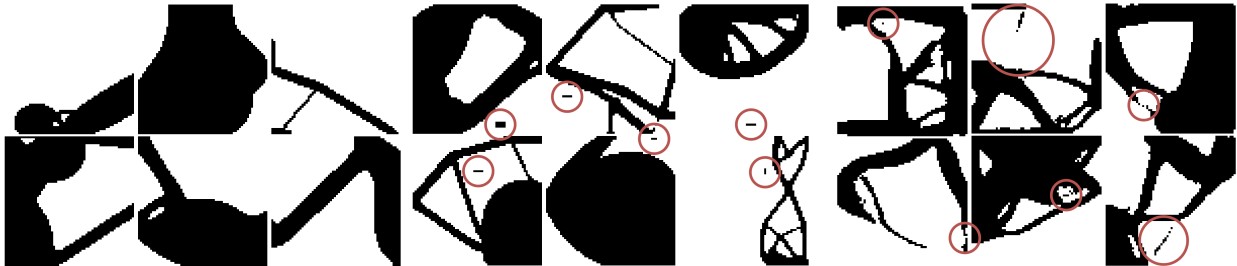

**(a)** Positive (connected) topologies **(b)** Procedurally-generated negatives **(c)** Rejection-sampled negatives

**Figure 10:** Topology optimization is a challenging structural engineering problem that searches for optimal placements of material. Valid (positive) structural topologies are completely continuous (left). Invalid (negative) topologies have floating material and can be procedurally-generated by artificially adding small floating patches of material (middle). They can also be collected through rejection-sampling of topologies created by a generative model (right). Constraint violation is annotated with red circles.

We train NDGMs using disconnected topologies as negative data, using the classification guidance dataset from Mazé & Ahmed (2023). The positive data is comprised of optimized, spatially continuous structures, while the negative data is largely comprised of procedurally-generated negatives with artificially-added floating components. For comparison, we create an alternative negative dataset by replacing procedurally-generated negatives in the dataset with rejection-sampled topologies generated by a vanilla GAN trained on the positive data. A simple continuity check flags any discontinuous topologies to add to the rejection-sampled negative dataset. A few topologies from each dataset are visualized in Figure 10. We hypothesize that these rejection-sampled negatives are 'harder' negatives (closer to the positive distribution) and are hence more informative than the procedurally-generated negatives.

**Table 6:** Means and standard deviations of performance metrics for the topology optimization problem over six tests with pairwise statistical significance comparisons. Lower scores are better. A model's symbol (✓/✗/†) is shown if statistically significant in a pairwise comparison. Best mean scores are **bolded**.

| Metric | Negative Dataset | GAN (baseline) | DO (sota) | MDD (ours) | DO † GAN ✗ | MDD ✓ GAN ✗ | MDD ✓ DO † |
|---|---|---|---|---|---|---|---|
| Invalidity | Procedural | 32.8±2.5 | 28.2±8.7 | **17.6±5.7** | – | ✓ | ✓ |
| Rate (%) | Rejection | 32.8±2.5 | 15.5±1.6 | **14.8±0.8** | † | ✓ | – |
| Violation | Procedural | **2.01±0.29** | 3.03±0.67 | 2.78±0.45 | ✗ | ✗ | – |
| Mag. $(10^{-3})$ | Rejection | 2.01±0.29 | 2.23±1.26 | **1.14±0.12** | – | ✓ | ✓ |
| Diversity | Procedural | 14.49±0.06 | 14.44±0.14 | **14.36±0.10** | – | ✓ | – |
| Score | Rejection | 14.49±0.06 | 14.52±0.13 | **14.38±0.10** | – | ✓ | ✓ |

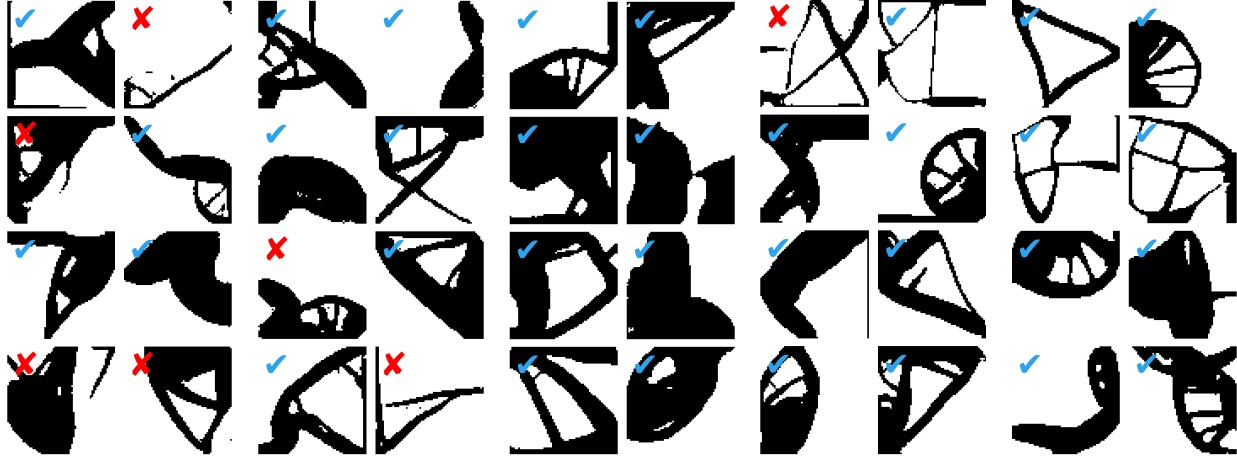

**(a)** GAN using no negatives   **(b)** GAN-DO using procedural negatives   **(c)** GAN-DO using rejected negatives   **(d)** GAN-MDD using procedural negatives   **(e)** GAN-MDD using rejected negatives

**Figure 11:** Visualization of topologies generated by a GAN trained only on positive topologies and GAN-DO/GAN-MDD models additionally trained on procedurally-generated negatives or rejection-sampled negatives. Valid samples are marked with a blue check mark while invalid samples are marked with a red X. Additional samples are visualized in Figures 15 through 19.

We run five types of experiments: GAN models trained on only positive data, GAN-DO and GAN-MDD trained on procedurally-generated data, and GAN-DO and GAN-MDD trained on rejection-sampled data. Some generated samples are visualized in Figure 11. We evaluate six instantiations of each experiment and evaluate pairwise comparisons using 2-sample t-tests with $p < 0.05$ significance. In evaluating models, we measure the proportion of generated topologies with disconnected components (invalidity rate), as well as the average fraction of image pixels disconnected from the largest continuous structure in each generated topology (violation magnitude). To identify mode collapse, we also measure DPP in the pixel space (diversity score). Numerical scores are presented in Table 6. These results suggest several noteworthy conclusions:

1. For both types of negative data, GAN-MDD outperforms GAN-DO in mean scores in every metric. This difference is statistically significant for invalidity rate score on procedurally-generated negatives and for violation magnitude score and diversity score on rejection-sampled negatives.

2. GAN-DO significantly outperforms the vanilla GAN in invalidity rate only for rejection-sampled negatives, suggesting that it is more sensitive to negative data quality than our GAN-MDD. As seen in Figure 12, GAN-DO can suffer from severe mode collapse. GAN-DO scores are also the most variable, indicating unpredictable training, stability, and convergence.

3. All NDGMs trained on rejection-sampled negatives achieve better mean scores in every metric than any NDGM trained on procedurally-generated negatives. This indicates that negative data quality can be even more impactful than the choice of NDGM type. It also highlights the potency of rejection sampling as a negative data sampling strategy when a black-box constraint check is available.

In summary, our benchmarking on the topology optimization problem illustrates the potency of GAN-MDD on a high-dimensional image-based problem but also illustrates the importance of negative data quality in NDGM performance. Best practices to generate high-quality negative data remain an open research area. Regardless, GAN-MDD is the highest overall performer on either type of negative data, particularly in constraint satisfaction rate.

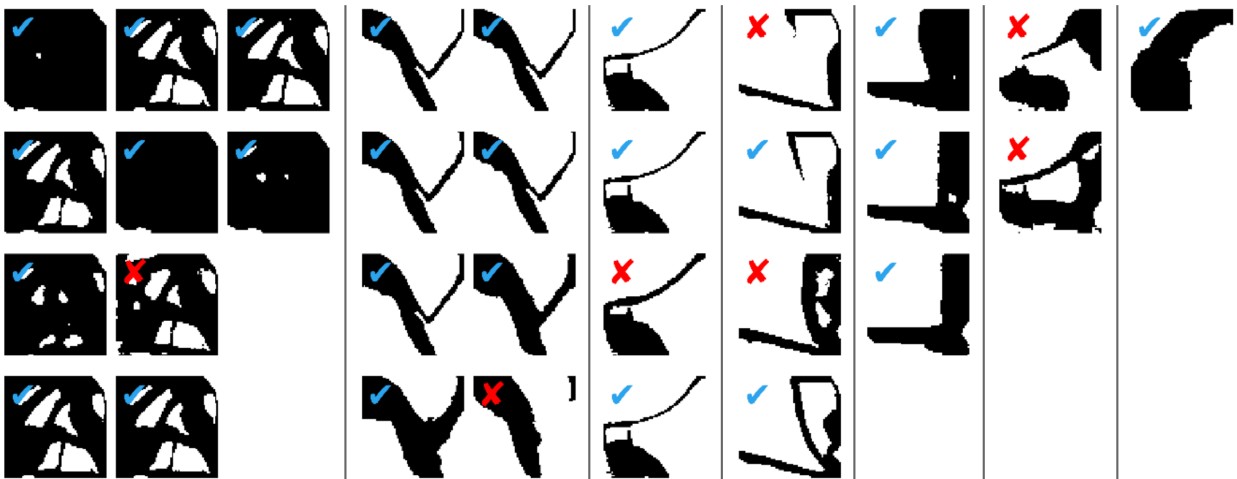

**Figure 12:** A batch of 32 random topologies generated by a select GAN-DO instantiation trained on rejection-sampled negatives. Samples fall under just a handful of data modes, indicating egregious mode collapse. A similar grouping is shown for a GAN-DO model trained on procedurally-generated negatives in Figure 20.

## 5   Discussion & Conclusion

**Adding pairwise density ratio estimation and diversity to NDGMs.** We presented a new NDGM formulation, GAN-MDD, which estimates individual density ratios, rather than conflating fakes and negatives as done in the current SOTA. Our model also incorporates a Determinantal Point Process-based diversity loss. These innovations empower GAN-MDD to achieve an optimal tradeoff between constraint satisfaction and distributional similarity, allowing it to outperform baseline models and existing NDGMs alike, across a variety of problems. In specific benchmarks, GAN-MDD manages to generate 1/6 as many constraint-violating samples as a vanilla model using only 1/8 as much data.

**GANs versus diffusion models using negative data.** Despite the growing popularity of diffusion models, GANs remain state of the art in many engineering design problems. On the challenging 2D problems, we find that: 1) vanilla DDPMs struggle to learn constraints and 2) DDPMs using guidance based on negative data can only achieve good constraint satisfaction at the expense of distributional similarity. This preliminary study indicates that DDPMs struggle to achieve as optimal of a tradeoff between constraint satisfaction and distributional similarity as our specialized adversarial NDGM model, at least in some problems. We look forward to future research which advances the capabilities of negative data diffusion models.

**NDGMs are underutilized.** We believe NDGMs are underutilized in engineering design. This assertion is substantiated by several observations: 1) The widespread use of vanilla models in engineering design despite their limitations (Regenwetter et al., 2022a). 2) The relatively low cost of collecting negative data versus positive data in many engineering contexts. 3) The overwhelming dominance of NDGMs over vanilla models in our engineering benchmarks. 4) The data-efficiency improvements we demonstrated using negative data.

**Generating high-quality negative data.** Selecting strategies to generate negative data is an important research question. In the final case study on topology optimization, rejection-sampling resulted in "stronger" negative data than the procedural generation method. It also required access to an oracle (constraint evaluator), which may be unavailable or prohibitively expensive in some applications. However, there are not always cheap, viable procedural generation approaches for negative data either. Effective negative data generation remains largely problem-dependent and the relative quality of negative data generation approaches is not necessarily straightforward. We anticipate that domain-agnostic methods to generate high-quality negative data could pair well with NDGMs and expand their impact.

**Leveraging negative data for classifier-based rejection sampling.** Negative data can be used to train supervised binary classification models to predict constraint satisfaction. These classifiers can then be applied during generative model inference to iteratively reject and regenerate any samples projected to violate constraints until a sufficiently large sample set is attained. If the generative model has a very low validity rate, this can increase the inference cost by orders of magnitude and add significant stochasticity to the generation process. To generate a single ship hull, for example, we estimate that a vanilla generative model would take a projected 33 times longer, or 222 times longer in a reasonable bad-case scenario, compared to an NDGM. Although expensive, this rejection sampling approach yields extremely strong validity rates, surpassing even the most accurate NDGMs. Though in many senses an apples-to-oranges juxtaposition, comparing NDGMs with rejection sampling offers hints about the theoretical limits of NDGM performance. Classifier-based rejection sampling using negative data is discussed in more detail in Appendix D.

**Limitations.** As we demonstrate, NDGMs are sensitive to the quality of their negative training data. Although negative data is often cheaper than positive data in engineering design problems, generating high-quality negative data may be challenging in some domains. In other domains, sourcing any kind of negative data may be impossible. In domains where high-quality negative data is unavailable, NDGMs will naturally be impractical.

## 6  Conclusion

In this paper, we presented a new negative-data generative model (NDGM), GAN-MDD, that innovates over previous methods by learning individual pairwise density ratios and avoiding mode-collapse using a diversity-based loss. In extensive benchmarks across constructed tests and a dozen real engineering problems, we demonstrated that GAN-MDD outperforms 10 other formulations. Moreover, it is data-efficient, generating 1/6 as many invalid samples as a vanilla model using 1/8 as much data. Finally, we showed that GAN-MDD can excel in challenging high-dimensional problems. GAN-MDD's data efficiency and optimal balance between distributional similarity and constraint satisfaction make it significantly more practical than existing generative models for engineering and design tasks. Thus, we anticipate and advocate for the more widespread use of NDGMs in data-driven engineering design and other constrained generative modeling domains.

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

# A Additional Visualization and Statistical Testing for Main Experiments

## A.1 2D Experiments

Generated distributions for all 10 baselines are included in Figures 13 and 14, alongside tabulated numerical scores in Table 7.

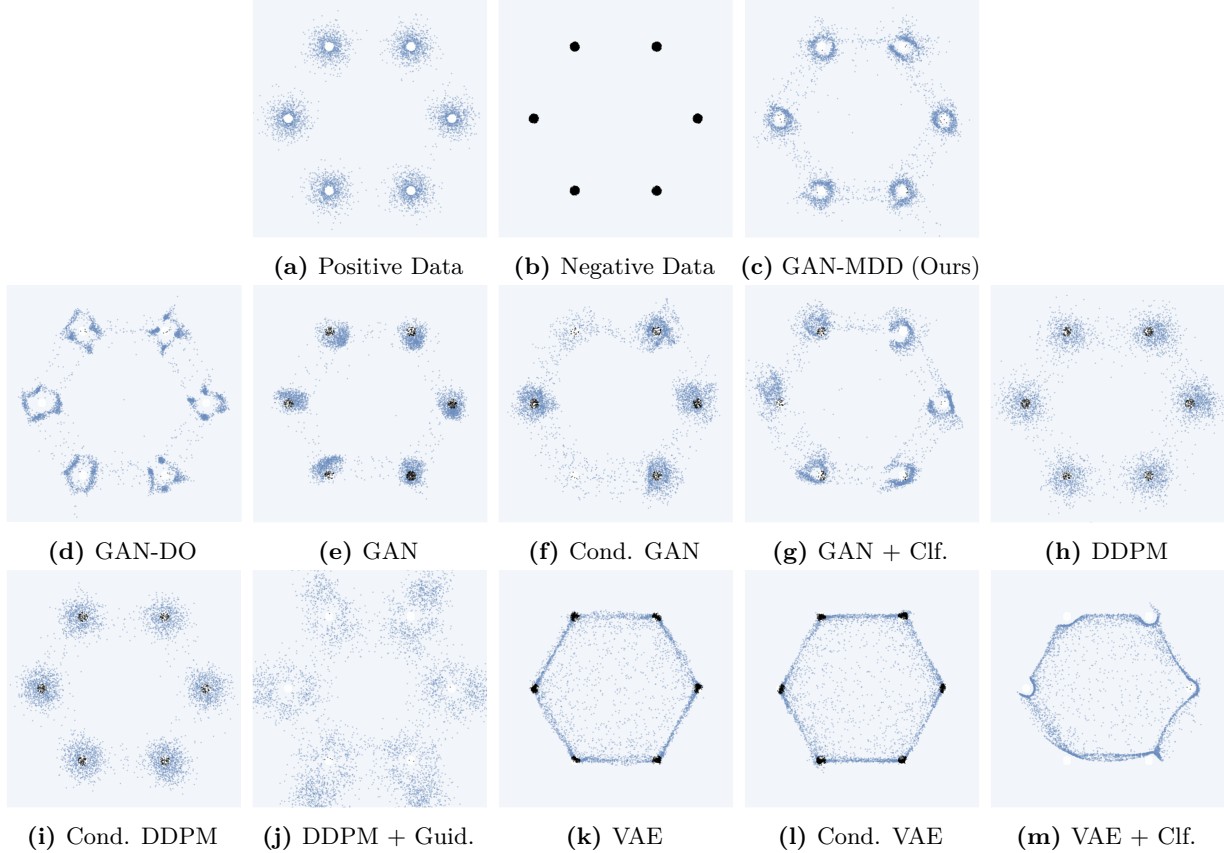

**(a)** Positive Data    **(b)** Negative Data    **(c)** GAN-MDD (Ours)

**(d)** GAN-DO    **(e)** GAN    **(f)** Cond. GAN    **(g)** GAN + Clf.    **(h)** DDPM

**(i)** Cond. DDPM    **(j)** DDPM + Guid.    **(k)** VAE    **(l)** Cond. VAE    **(m)** VAE + Clf.

**Figure 13:** Visual comparison of our GAN-MDD against all 10 baseline models on `Problem 1`. Positive data points and samples are shown in blue and negative ones in black.

|  | Problem 1 | | | Problem 2 | | |
|---|---|---|---|---|---|---|
|  | Inval. (%) (↓) | F1 (↑) | Diversity (↓) | Inval. (%) (↓) | F1 (↑) | Diversity (↓) |
| VAE | 25.51±0.57 | 0.397±0.017 | 15.01±0.02 | 14.19±0.58 | 0.751±0.007 | 14.36±0.02 |
| VAE-Cond | 26.64±1.05 | 0.389±0.009 | 15.05±0.01 | 14.23±0.44 | 0.755±0.003 | 14.36±0.01 |
| VAE-Clf | 0.05±0.06 | 0.458±0.012 | 15.05±0.04 | 0.06±0.13 | 0.345±0.072 | 15.42±0.19 |
| DDPM | 8.60±0.65 | 0.820±0.021 | 14.86±0.02 | 14.97±0.68 | 0.848±0.005 | 13.92±0.04 |
| DDPM-Cond | 8.30±0.55 | 0.814±0.020 | 14.94±0.01 | 11.44±0.81 | 0.858±0.005 | 13.98±0.02 |
| DDPM-Guid | 0.26±0.04 | 0.404±0.014 | 14.07±0.03 | 5.92±0.54 | 0.821±0.017 | 14.10±0.02 |
| GAN | 8.32±3.23 | 0.596±0.109 | 15.17±0.08 | 4.39±0.55 | 0.859±0.007 | 14.09±0.02 |
| GAN-Cond | 8.13±1.02 | 0.725±0.075 | 15.06±0.03 | 5.24±1.18 | 0.850±0.008 | 14.05±0.07 |
| GAN-Clf | 1.06±1.14 | 0.609±0.139 | 15.14±0.11 | 2.22±0.94 | 0.749±0.101 | 14.30±0.20 |
| GAN-DO | 0.25±0.24 | 0.493±0.054 | 15.11±0.11 | 0.52±0.06 | 0.754±0.018 | 14.34±0.03 |
| **GAN-MDD (ours)** | 0.40±0.13 | 0.700±0.028 | 14.99±0.05 | 0.73±0.19 | 0.830±0.008 | 14.16±0.02 |
| GAN-MDD Rank | 4 | 4 | 4 | 3 | 5 | 6 |

**Table 7:** Mean and standard deviations for 2D problem scores over six instantiations. Model scores that beat or are beaten by GAN-MDD's scores to $p < 0.05$ significance are colored light blue or dark red, respectively. GAN-MDD's mean score ranking is included in the last row.

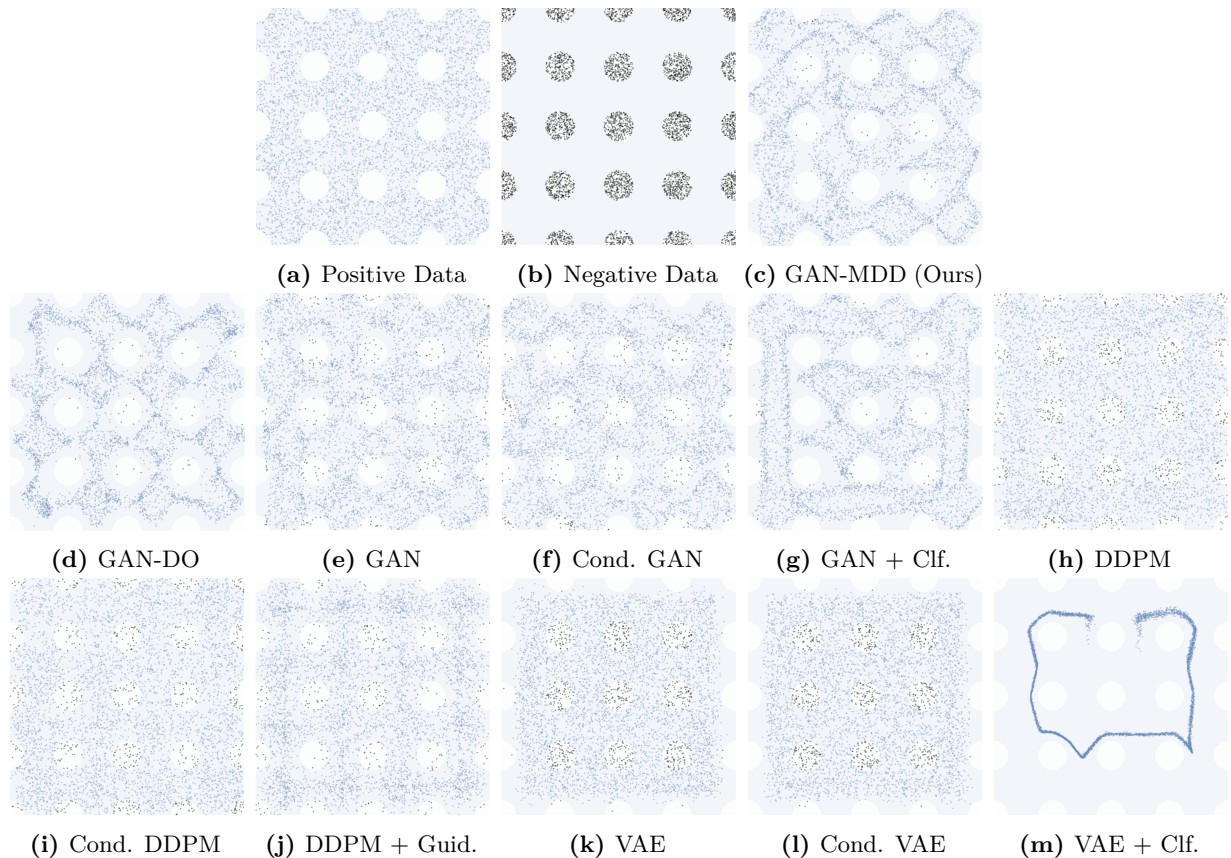

**Figure 14:** Visual comparison of our GAN-MDD against all 10 baseline models on `Problem` 2. Positive data points and samples are shown in blue and negative ones in black.

## A.2 Dataset Size

Full scores with standard deviations and significnace testing are included in Table 8.

**Table 8:** Full scores from the dataset size study presented in the main paper, showing both mean scores and standard deviations over the four runs. Scores significantly ($p < 0.05$ using a 2-sample t-test) better than the corresponding GAN trained with the same amount of positive data are bolded. **Lower is better.** The relatively large deviation seen in vanilla models may explain the slight counterintuitive increase in invalidity rate with more data in `Problem` 1.

| (a) Models | Negative Samples | (b) Problem 1 Positive Samples 1K | 4K | 16K | (c) Problem 2 Positive Samples 1K | 4K | 16K |
|---|---|---|---|---|---|---|---|
| GAN | 0 | $10.7\% \pm 5.3\%$ | $11.6\% \pm 2.9\%$ | $11.9\% \pm 1.6\%$ | $6.0\% \pm 1.5\%$ | $3.2\% \pm 1.3\%$ | $3.3\% \pm 0.7\%$ |
| GAN-MDD | 1K | $\mathbf{2.0\% \pm 2.3\%}$ | $\mathbf{0.7\% \pm 0.2\%}$ | $\mathbf{1.5\% \pm 0.8\%}$ | $\mathbf{2.3\% \pm 0.7\%}$ | $1.6\% \pm 0.7\%$ | $\mathbf{1.9\% \pm 0.7\%}$ |
| GAN-MDD | 4K | $\mathbf{0.5\% \pm 0.3\%}$ | $\mathbf{0.5\% \pm 0.3\%}$ | $\mathbf{0.5\% \pm 0.2\%}$ | $\mathbf{0.6\% \pm 0.2\%}$ | $\mathbf{0.7\% \pm 0.3\%}$ | $\mathbf{0.7\% \pm 0.2\%}$ |
| GAN-MDD | 16K | $\mathbf{0.5\% \pm 0.3\%}$ | $\mathbf{0.7\% \pm 0.5\%}$ | $\mathbf{0.6\% \pm 0.3\%}$ | $\mathbf{0.5\% \pm 0.1\%}$ | $\mathbf{0.2\% \pm 0.1\%}$ | $\mathbf{0.2\% \pm 0.1\%}$ |

### A.3 Diversity Loss Ablation Studies

Full scores for diversity loss studies are included in Table 9.

| | Problem 1 | | | Problem 2 | | |
|---|---|---|---|---|---|---|
| | Inval. (%) (↓) | F1(↑) | Diversity (↓) | Inval. (%) (↓) | F1 (↑) | Diversity (↓) |
| GAN-DO ($\gamma = 0.0$) | 0.25±0.24 | 0.493±0.054 | 15.111±0.110 | 0.52±0.06 | 0.754±0.017 | 14.343±0.027 |
| GAN-DO ($\gamma = 0.1$) | 0.63±0.26 | 0.629±0.013 | 14.854±0.019 | 1.36±0.15 | 0.817±0.010 | 14.201±0.011 |
| GAN-DO ($\gamma = 0.2$) | 0.80±0.14 | 0.645±0.014 | 14.766±0.020 | 2.94±0.29 | 0.836±0.004 | 14.097±0.015 |
| GAN-MDD ($\gamma = 0.0$) | 0.33±0.27 | 0.627±0.159 | 15.119±0.178 | 0.84±0.38 | 0.829±0.010 | 14.165±0.024 |
| GAN-MDD ($\gamma = 0.1$) | 0.40±0.13 | 0.700±0.028 | 14.988±0.045 | 0.73±0.19 | 0.830±0.008 | 14.159±0.016 |
| GAN-MDD ($\gamma = 0.2$) | 0.55±0.15 | 0.794±0.013 | 14.716±0.010 | 2.01±0.28 | 0.858±0.008 | 14.043±0.011 |

**Table 9:** Scores for GAN-DO and GAN-MDD on `Problem 1` and `Problem 2` over a variety of diversity loss weights. Mean scores and standard deviations over six instantiations are shown.

### A.4 Topology Optimization

We visualize more samples generated by GAN, GAN-DO and GAN-MDD, annotating validity in Figures 15 through 19 and visualize mode collapse in GAN-DO on the synthetic negative data in Figure 20.

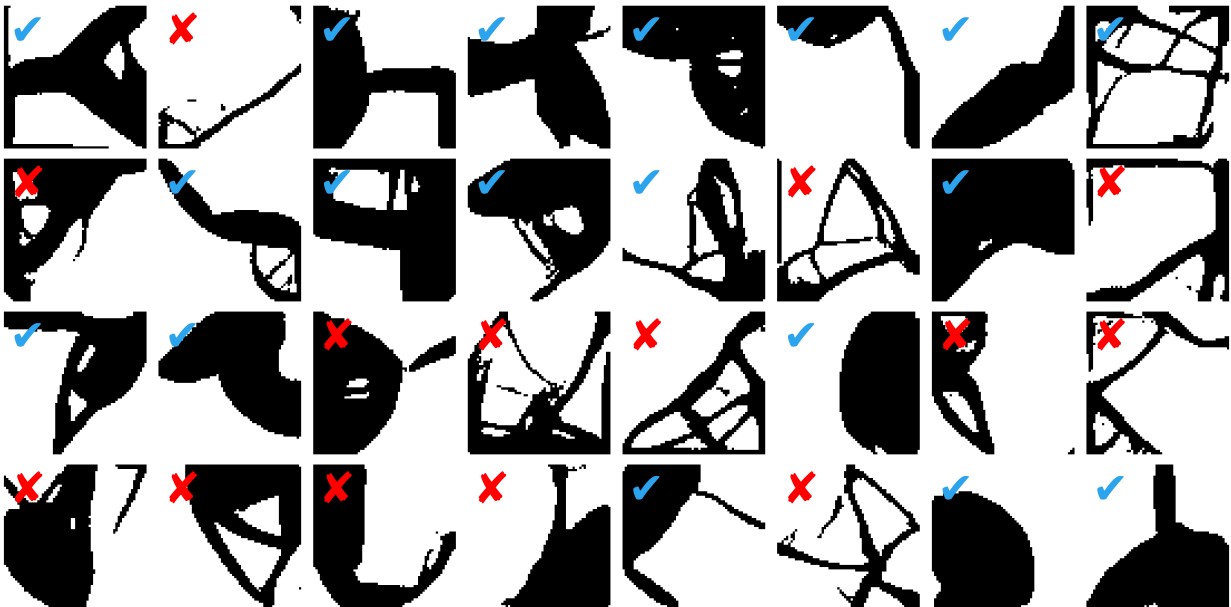

**Figure 15:** Randomly-selected topologies generated by GAN with constraint violations annotated.

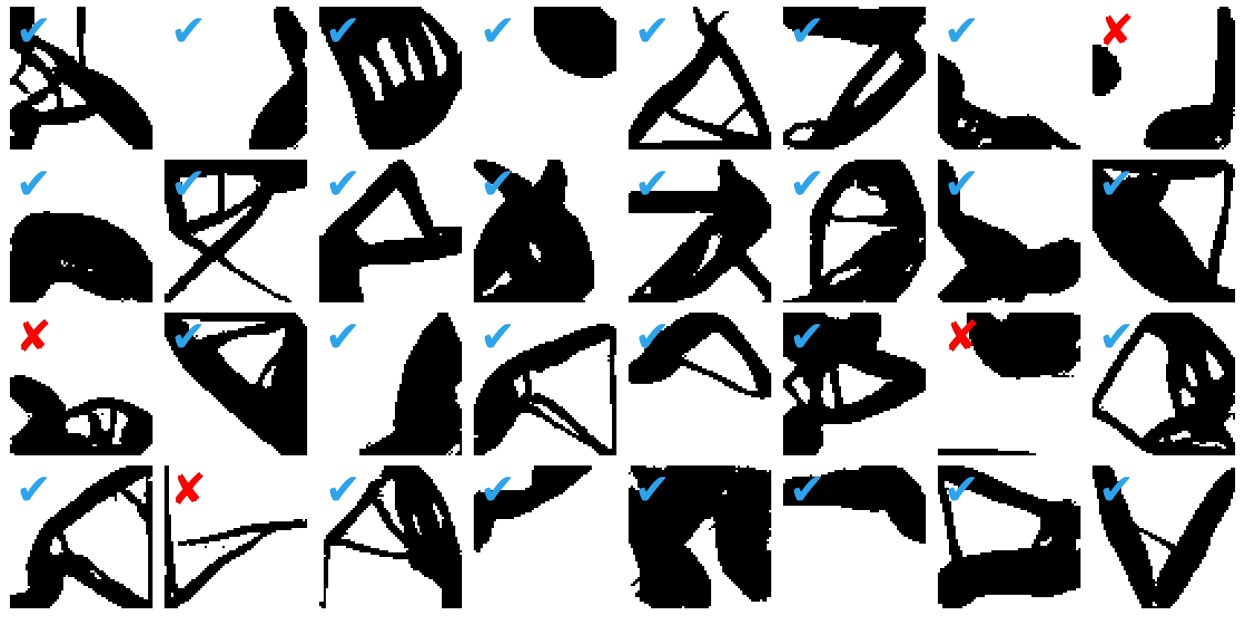

**Figure 16:** Randomly-selected topologies generated by GAN-DO trained on procedurally-generated negative data with constraint violations annotated.

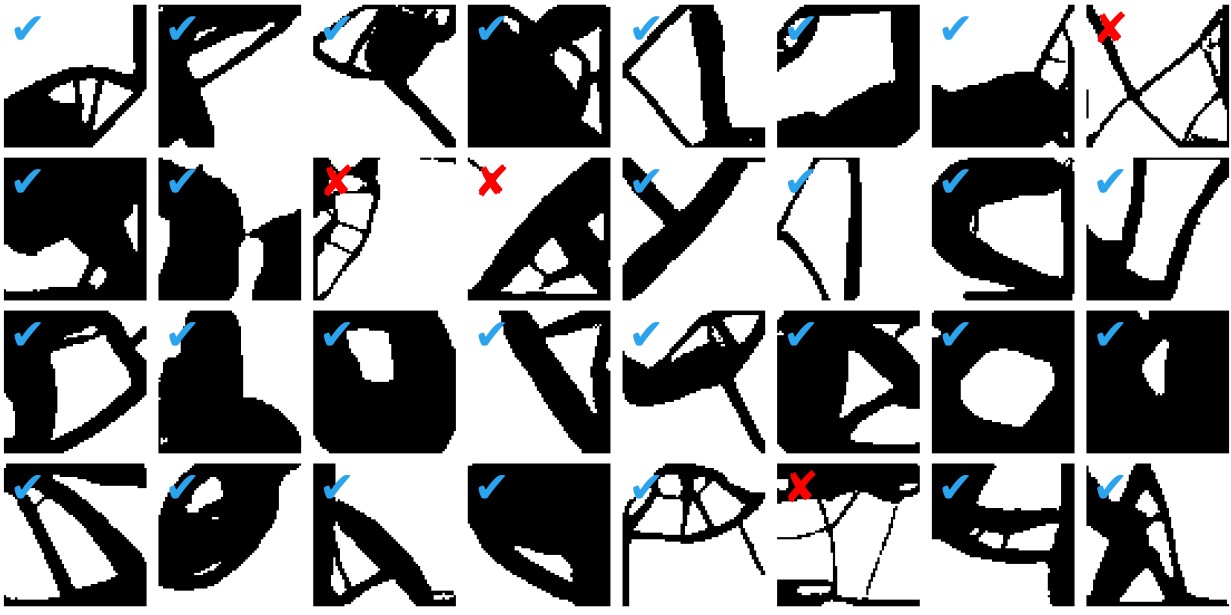

**Figure 17:** Randomly-selected topologies generated by GAN-DO trained on rejection-sampled negative data with constraint violations annotated.

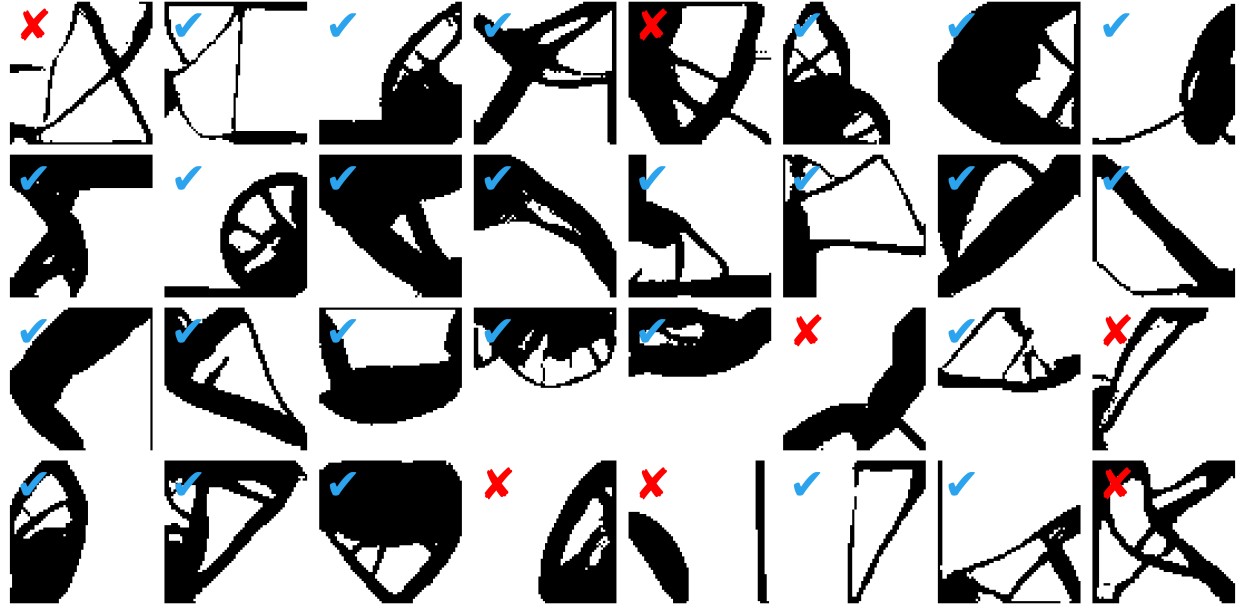

**Figure 18:** Randomly-selected topologies generated by GAN-MDD trained on procedurally-generated negative data with constraint violations annotated.

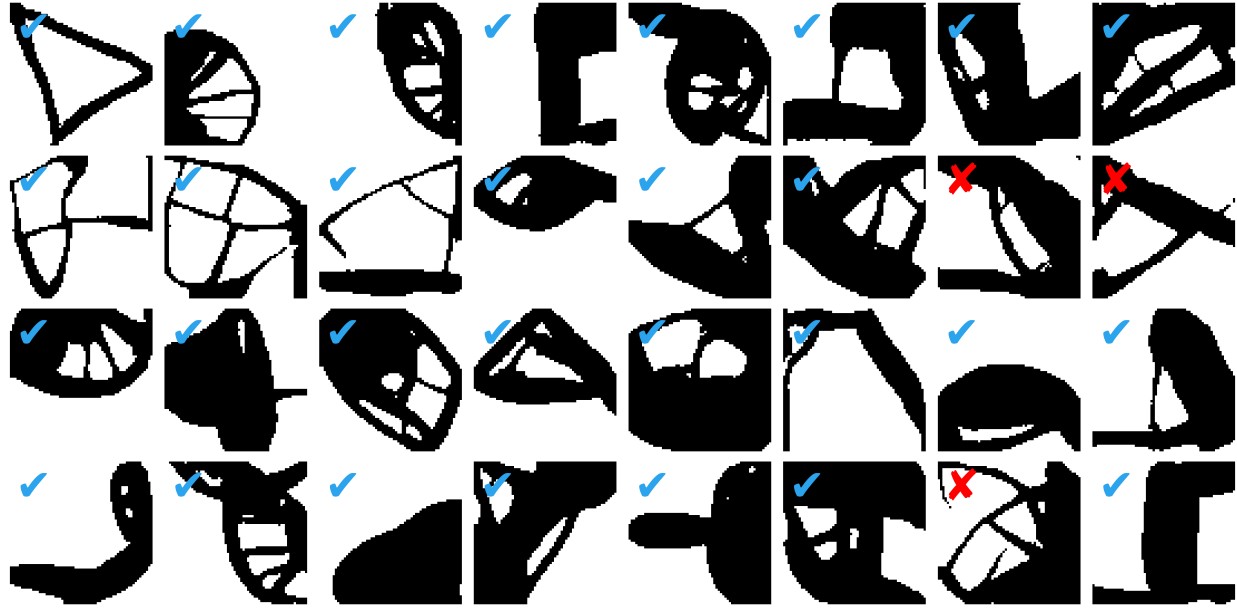

**Figure 19:** Randomly-selected topologies generated by GAN-MDD trained on rejection-sampled negative data with constraint violations annotated.

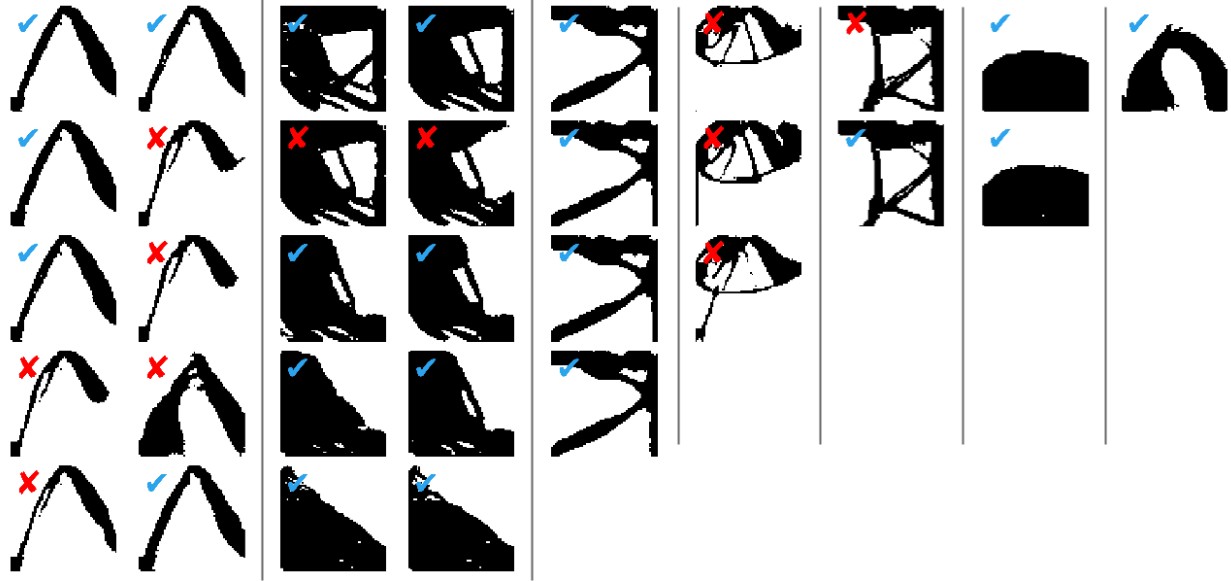

**Figure 20:** A batch of 32 random topologies generated by GAN-DO trained on procedurally-generated negatives. Topologies are manually grouped into just a handful of data modes, indicating severe mode collapse.

## B  Pseudocode

Pseudocode for our GAN-MDD training formulation is shown below:

---
**Algorithm 1** GAN-MC Training Procedure
---
**while** $step \leq n_{steps}$ **do**
    Sample $P_{batch} \sim P_{dataset}$ and $N_{batch} \sim N_{dataset}$
    Sample $\epsilon \sim N(0,1)$
    $G_{batch} = \text{Generator}(\epsilon)$
    $D^P_{preds} = \text{Discriminator}(P_{batch})$
    $D^N_{preds} = \text{Discriminator}(N_{batch})$
    $D^G_{preds} = \text{Discriminator}(G_{batch})$
    loss_fn = CategoricalCrossEntropy()
    $D\_loss = \text{loss\_fn}(D^G_{preds}, 0) + \text{loss\_fn}(D^P_{preds}, 1) + \text{loss\_fn}(D^N_{preds}, 2)$
    $Diversity\_loss = \text{DPP}(G_{batch})$
    $G\_loss = \text{loss\_fn}(D^G_{preds}, 1) + \gamma \cdot Diversity\_loss$
    Optimize(Discriminator, $D\_loss$)
    Optimize(Generator, $G\_loss$)
    $step = step + 1$
**end while**

---

## C  Extended Background and Related Work

**Constraints in Engineering Problems.** Generally, we can categorize the constraint information of engineering problems into four types (Regenwetter et al., 2023).

   (i)  No Constraint Information: No information about constraints is given or can be collected, and learning constraints is typically infeasible or extremely challenging in a finite data regime.

  (ii)  'Negative' Dataset of Invalid Designs: A collection of constraint-violating negative designs is available. Our method leverages such negative data to learn a constraint-satisfying generative model.

 (iii)  Constraint Check: A black-box 'oracle' is available to determine whether a design satisfies constraints. This check may be computationally expensive, limiting its use.

 (iv)  Closed-form Constraints: An inexpensive closed-form constraint is available. In such scenarios, direct optimization is often favored over generative models in design problems. In other cases, constraint-enforcing rules can be built into the model structure, an approach used in some generative models for molecular design (Cheng et al., 2021; Imrie et al., 2020).

We note that each level of constraint information is strictly more informative than the previous. In this paper, we focus on the scenario in which a limited dataset of negative samples is available (ii) or can be generated using an oracle (iii), but closed-form constraints are not available. This scenario is common in applications such as structural design, mobility design (e.g., cars, bikes, ships, airplanes), and material synthesis.

**Density Ratio Estimation.** Density Ratio Estimation (DRE) (Sugiyama et al., 2012b) is a critical technique in machine learning, particularly when evaluating distributions is infeasible or is computationally expensive (Mohamed & Lakshminarayanan, 2016). DRE techniques are heavily employed for generative modeling and score matching estimation (Goodfellow et al., 2014; Gutmann & Hyvärinen, 2010; Srivastava et al., 2023; Choi et al., 2022). In the context of GANs (Goodfellow et al., 2014; Arjovsky et al., 2017), the DRE methodology forms the underlying basis for their operation. A well-known technique for DRE is probabilistic classification Sugiyama et al. (2012b), where a binary classifier is used to learn the ratio. However, accurately performing DRE with finite samples is particularly challenging in high-dimensional spaces. To overcome this challenge, prior works have employed a divide-and-conquer approach. An example of this is the Telescoping Density Ratio Estimation (TRE) method (Gutmann & Hyvärinen, 2010; Rhodes et al., 2020), which divides the problem into a sequence of easier DRE sub-problems. Despite its success, there are limitations to this approach, especially when the number of intermediate bridge distributions is increased.

Noise contrastive estimator (NCE (Gutmann & Hyvärinen, 2010)) and hybrid generative models (Srivastava et al., 2023; 2017; Rhodes et al., 2020) are also based on the density ratio as underlying methodology, providing a flexible paradigm for large scale generative modeling.

**Density Ratio Estimation in the Negative Data Context.** Let $p_n$ denote the negative distribution i.e., the distribution of constraint-violating datapoints. Instead of training using only the positive distribution $p_p$, NDGM formulations seek to train a generative model $p_\theta$ using both $p_p$ and $p_n$. Assuming mutual absolute continuity of $p_p, p_\theta$ and $p_n$, and starting from first principles, we can now re-write Eq. 2 as:

$$
\begin{aligned}
&\arg\min_\theta \int p_\theta(\mathbf{x}) \left[\log p_\theta(\mathbf{x}) - \log p_p(\mathbf{x})\right] d\mathbf{x} \\
&= \arg\min_\theta \int p_\theta(\mathbf{x}) \left[\log p_\theta(\mathbf{x}) - \log p_p(\mathbf{x}) + \left(\log \frac{p_n(\mathbf{x})}{p_n(\mathbf{x})}\right)\right] d\mathbf{x} \\
&= \arg\min_\theta \int p_\theta(\mathbf{x}) \left[\log \frac{p_\theta(\mathbf{x})}{p_n(\mathbf{x})} - \log \frac{p_p(\mathbf{x})}{p_n(\mathbf{x})}\right] d\mathbf{x}.
\end{aligned} \tag{13}
$$

While the solution for Eq. 13 is the same as the solution for Eq. 2 i.e. $p_{\theta^*} = p_p$, the model is now directly incentivized to allocate the same amount of probability mass to the samples from $p_n$ as does the data distribution $p_p$. This ensures that when trained using finite $N$, the model avoids allocating high probability mass to invalid samples. In other words, training under Eq. 13 encourages the model to minimize its discrepancy with respect to $p_p$ such that its discrepancy with respect to $p_n$ matches exactly that of $p_p$ and $p_n$.

**Generative Models for Engineering Design.** Generative models have recently seen extensive use in design generation tasks (Regenwetter et al., 2022a). Generative Adversarial Nets, for example, have seen extensive use in many applications. In topology optimization, GANs (Li et al., 2019; Rawat & Shen, 2019; Oh et al., 2018; 2019; Sharpe & Seepersad, 2019; Nie et al., 2021; Yu et al., 2019; Valdez et al., 2021; Mazé & Ahmed, 2023) are often used to create optimal topologies, potentially bypassing time-consuming iterative solvers. In computational materials design, GANs (Tan et al., 2020; Yang et al., 2018; Zhang et al., 2021; Mosser et al., 2017; Lee et al., 2021; Liu et al., 2019), VAEs (Cang et al., 2018; Li et al., 2020; Liu et al., 2020; Wang et al., 2020; Xue et al., 2020; Tang et al., 2020; Chen & Liu, 2021), and other models are used to generate synthetic data to better learn process-structure-property relations (Bostanabad et al., 2018). A variety of generative models have been applied to 2D shape synthesis problems (Yilmaz & German, 2020; Chen & Fuge, 2018; Chen et al., 2019; Chen & Fuge, 2019; Nobari et al., 2022; Li et al., 2021; Dering et al., 2018), such as airfoil design, and 3D shape synthesis problems (Shu et al., 2020; Nobari et al., 2022; Brock et al., 2016; Zhang et al., 2019) such as mechanical component synthesis in engineering design. Finally, generative models have been proposed as a method to tackle various miscellaneous product and machine design tasks (Deshpande & Purwar, 2019; Sharma & Purwar, 2020; Regenwetter et al., 2021; Deshpande & Purwar, 2020).

**Constraint Satisfaction in Machine Learning.** From a general point of view, Constraint Satisfaction Problems (CSPs) have been long studied in computer science and optimization about optimal allocation, graph search, games, and path planning (Russell, 2010). However, such constraints are mostly related to algorithmic complexity and memory allocation. In generative design, the goal of constraint satisfaction differs as it aims to achieve both high-performance designs and diverse distribution coverage through probabilistic modeling (Regenwetter et al., 2022a). Recently, Neural Constraint Satisfaction (Chang et al., 2023) has been proposed to deal with objects in a scene to solve intuitive physics problems (Smith et al., 2019; Hamrick et al., 2018). In the CAD domain, structured models to handle constraints have been proposed (Seff et al., 2021; Para et al., 2021). Conditional generative models have been proposed for structural topology optimization (Nie et al., 2021), leveraging physical fields (Nie et al., 2021; Mazé & Ahmed, 2023), dense approximations (Giannone & Ahmed, 2023), and trajectory alignment (Giannone et al., 2024) for high-quality candidate generation. These approaches rely on explicit constraint satisfaction. Instead, we focus on implicit constraint satisfaction, leveraging a dataset of invalid configurations to enhance the model capacity to generate valid designs.

## D Comparison to Rejection Sampling

In constrained generation problems, rejection sampling is a simple, yet powerful strategy to ensure constraint satisfaction. Unfortunately, a black-box constraint check is not always available and may be prohibitively costly. In the negative data domain, negative data provides an opportunity to train a supervised constraint evaluation surrogate which can be used during rejection sampling, generally at lower cost than a query to a ground-truth constraint oracle. To generate a batch of samples, the generative model and predictive model will be queried in a loop, discarding any predicted invalid samples and collecting predicted valid samples until a sufficient number of predicted valid samples is accumulated.

### D.1 Benchmarking

We benchmark a vanilla GAN, VAE, and DDPM in this manner, training a supervised classifier using the negative data, then applying the classifier during inference. Figure 21 and Table 10 present the scores, while Figures 22 and 23 show the distribution plots. Rejection sampling outperforms GAN-MDD and GAN-DO (and most baselines) in terms of invalidity rate. Noting that a supervised classifier has a much simpler task than a constrained generative model, we would not expect NDGMs to learn constraints boundaries more precisely than classifiers. Thus, it is unrealistic to expect NDGMs to beat this rejection sampling baseline in constraint satisfaction scores. As expected, vanilla models augmented by rejection sampling still often underperform NDGMs in F1 score.

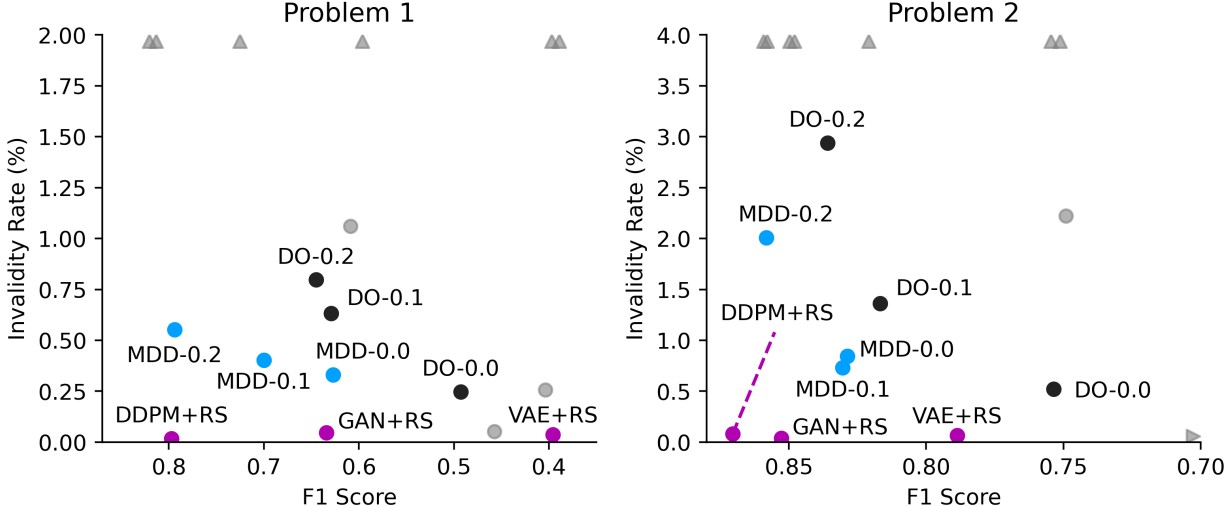

**Figure 21:** Comparison of F1 scores (↑) and invalidity rates (↓) for GAN-MDD and GAN-DO against rejection sampling baselines ("+RS") on `Problem 1` (left) and `Problem 2` (right). Gray markers indicate scores for other models benchmarked in the main paper. Mean scores over six instantiations are plotted. Scores closer to the bottom left are more optimal. Triangular markers indicate that the score lies off the plot in the indicated direction.

|  | Problem 1 | | | Problem 2 | | |
|---|---|---|---|---|---|---|
|  | **Inval. (%) (↓)** | **F1 Score (↑)** | **Diversity (↓)** | **Inval. (%) (↓)** | **F1 Score (↑)** | **Diversity (↓)** |
| VAE+RS | 0.037±0.042 | 0.396±0.007 | 14.976±0.010 | 0.067±0.032 | 0.789±0.002 | 14.393±0.004 |
| GAN+RS | 0.047±0.055 | 0.634±0.050 | 15.071±0.060 | **0.037±0.018** | 0.853±0.015 | 14.157±0.031 |
| DDPM+RS | **0.017±0.021** | **0.797±0.006** | **14.829±0.013** | 0.080±0.059 | **0.870±0.005** | **14.086±0.017** |

**Table 10:** Detailed scores for vanilla models with rejection sampling on `Problem 1` and `Problem 2`. Best scores are bolded. Mean scores and standard deviations over six instantiations are shown.

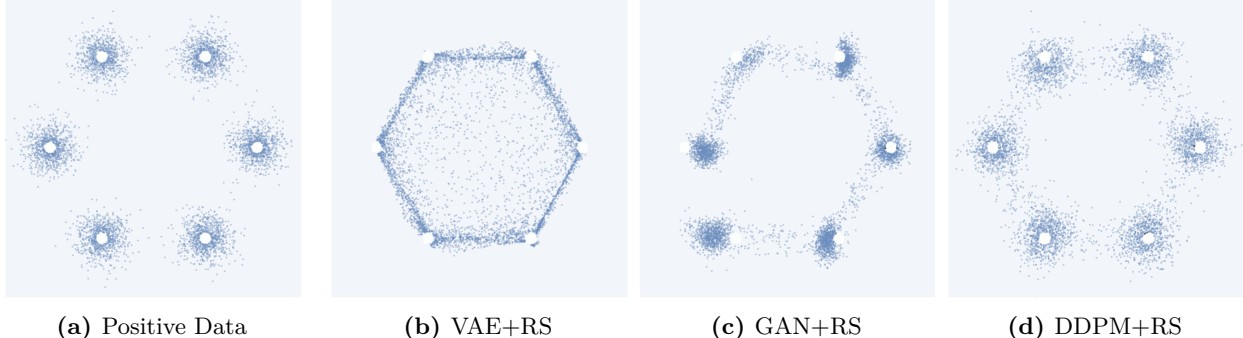

**(a)** Positive Data      **(b)** VAE+RS      **(c)** GAN+RS      **(d)** DDPM+RS

**Figure 22:** Generated distributions for vanilla models on `Problem 1` after rejection sampling using a classifier trained to distinguish negative and positive data.

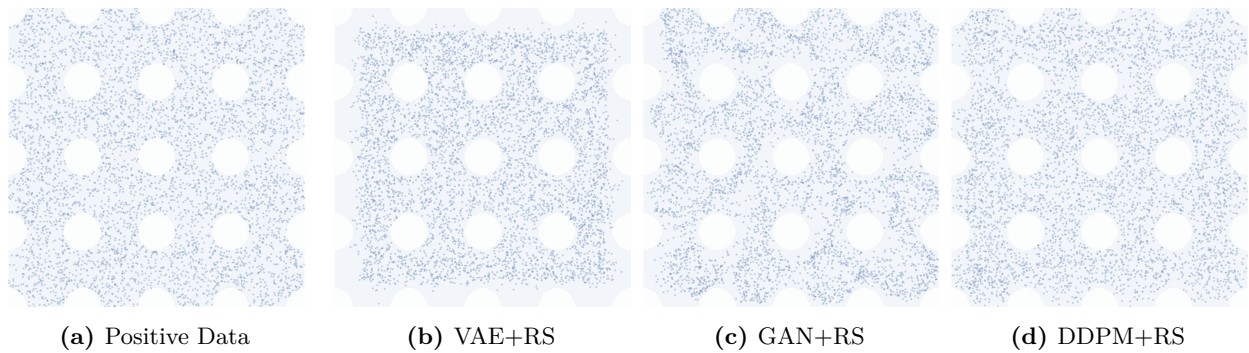

**(a)** Positive Data      **(b)** VAE+RS      **(c)** GAN+RS      **(d)** DDPM+RS

**Figure 23:** Generated distributions for vanilla models on `Problem 2` after rejection sampling using a classifier trained to classify negative vs. positive data.

### D.2 Rejection Sampling Carries Significant Inference-Time Costs

Rejection sampling using a predictive model carries significant downsides when deployed in practice. Most notably, the number of model calls is expected to at least double, and it may increase by orders of magnitude if the generative model's validity rate is very low. Furthermore, the inference time of the model will be stochastic. For a model with validity rate $v$, the expected mean and variance of the number of function calls ($n_c$) is:

$$\mathbb{E}[n_c] = \frac{2}{v}; \ Var[n_c] = \frac{4(1-v)}{v^2} \tag{14}$$

To illustrate the huge inference cost scaling and unpredictability for models with low validity rate, Table 11 compares the rejection sampling cost of a GAN trained on `Problem 1` and a GAN trained on the ship hull engineering problem. The validity rate of these two models is hugely different – 91.7% and 6.1%, respectively. Rejection sampling increases the expected number of model calls from 1 to 2.18 for `Problem 1`, and to 33.17 for the ship hull problem. We also estimate the expected worst case scenario out of 1000 inference runs. As shown, in one of a thousand inferences, we expect to require at least 222 calls to the model to generate a valid ship hull.

Inference memory costs also increase, further complicating real-time deployment of generative models. This combination of factors can make rejection sampling less practical for real-time deployment of generative models. These significant downsides motivate the further exploration and development of NDGMs.

Naturally, NDGMs can also be used with rejection sampling to accelerate inference due to their higher validity rates and sometimes superior distribution learning. Nonetheless, rejection sampling serves as a useful reference point for projecting the frontier of NDGM modeling capabilities, assuming perfect generation based on the quality of constraint boundary estimates learned by supervised models.

|                                      | Problem 1 | Ship Hull |
| ------------------------------------ | --------- | --------- |
| GAN model validity rate (%)          | 91.68     | 6.07      |
| Expected number of model calls       | 2.182     | 33.17     |
| Variance in number of model calls    | 0.396     | 4081      |
| 99.9th percentile number of model calls | 6      | 222       |

**Table 11:** Inference statistics for `Problem 1` and ship hull. As seen for the ship hull design problem, low model validity rates cause significantly higher unpredictability and expected number of model calls.

## E   Double Discriminator Training Formulations

### E.1   Formulation

Though using GAN-MDD's multi-class discriminator to learn pairwise density ratios between $p_p$, $p_n$, and $p_\theta$ is simpler, we can also accomplish the task using multiple discriminative models. For example, $f_\phi$ can estimate the ratio $p_p/p_\theta$ (a standard GAN discriminator), while $f_\psi$ estimates $p_n/p_\theta$. The loss is then expressed as:

$$
\begin{aligned}
\mathcal{L}(\theta, \phi, \psi) = {} & \mathbb{E}_{p_p(\mathbf{x})}[\log f_\phi(\mathbf{x})] + \mathbb{E}_{p_\theta(\mathbf{x})}[1 - \log(f_\phi(\mathbf{x}_\theta))] \\
& - \lambda \mathbb{E}_{p_n(\mathbf{x})}[\log f_\psi(\mathbf{x})] - \lambda \mathbb{E}_{p_\theta(\mathbf{x})}[1 - \log(f_\psi(\mathbf{x}_\theta))].
\end{aligned}
\tag{15}
$$

Here, $\lambda \in [0, 1]$ is a tuning parameter adjusting the weight of the negative data's contribution to the loss and avoiding instability. Optimal discriminators learn:

$$
f_\phi(\mathbf{x}) = \frac{p_p(\mathbf{x})}{(p_\theta(\mathbf{x}) + p_p(\mathbf{x}))}, \quad f_\psi(\mathbf{x}) = \frac{p_n(\mathbf{x})}{(p_\theta(\mathbf{x}) + p_n(\mathbf{x}))}.
\tag{16}
$$

The rationale behind the double discriminator algorithm is intuitive when viewed as an expansion of a vanilla GAN. The generator aims for its samples to be classified as positive by the original discriminator and not as negative by the extra discriminator.

We benchmark this double discriminator variant below, titled GAN-DDD (double discriminator + diversity). In practice, we also find that an alternate formulation that combines this simple two-discriminator concept with discriminator overloading (DO) also works well in many cases. The alternative formulation consists of the classic discriminator, $f_\phi$ estimating $p_p/p_\theta$ and an overloaded discriminator, $f_\psi$ estimating $(p_p + p_\theta)/p_n$. The total loss function is then expressed as:

$$
\begin{aligned}
\mathcal{L}(\theta, \phi, \psi) = {} & \mathbb{E}_{p_p(\mathbf{x})}[\log f_\phi(\mathbf{x})] + \mathbb{E}_{p_\theta(\mathbf{x})}[1 - \log(f_\phi(\mathbf{x}_\theta))] \\
& + \lambda \mathbb{E}_{p_p(\mathbf{x})}[\log f_\psi(\mathbf{x})] + \lambda \mathbb{E}_{p_\theta(\mathbf{x})}[\log f_\psi(\mathbf{x}_\theta)] + \lambda \mathbb{E}_{p_n(\mathbf{x})}[1 - \log(f_\psi(\mathbf{x}))].
\end{aligned}
\tag{17}
$$

Once again, $\lambda$ is a weighting parameter modulating the contribution of the negative data.

### E.2   Testing

We benchmark GAN-DDD on the 2D test problems using a negative data weight of $\lambda = 0.4$. Scores are plotted against other models in Figure 24 and tabulated in Table 12, while generated distributions are plotted in Figure 25. Although it performs very well in `Problem 1`, GAN-DDD generates many invalid samples in `Problem 2`. In general, we find that the careful tuning of the negative data weighting parameter and diversity loss weighting parameter ($\lambda$ and $\gamma$, respectively) have significant impacts on the tradeoff between distributional similarity and validity. Therefore, we generally recommend GAN-MDD as a simpler off-the-shelf method, even though GAN-DDD may perform better in specific scenarios.

| | Problem 1 | | | Problem 2 | | |
|---|---|---|---|---|---|---|
| | Inval. (%) (↓) | F1(↑) | Diversity (↓) | Inval. (%) (↓) | F1 (↑) | Diversity (↓) |
| GAN-DDD | 0.60±0.16 | 0.793±0.015 | 14.703±0.026 | 5.24±0.56 | 0.858±0.004 | 14.028±0.036 |

**Table 12:** Scores for GAN-DDD on `Problem 1` and `Problem 2`. Mean scores and standard deviations over six instantiations are shown.

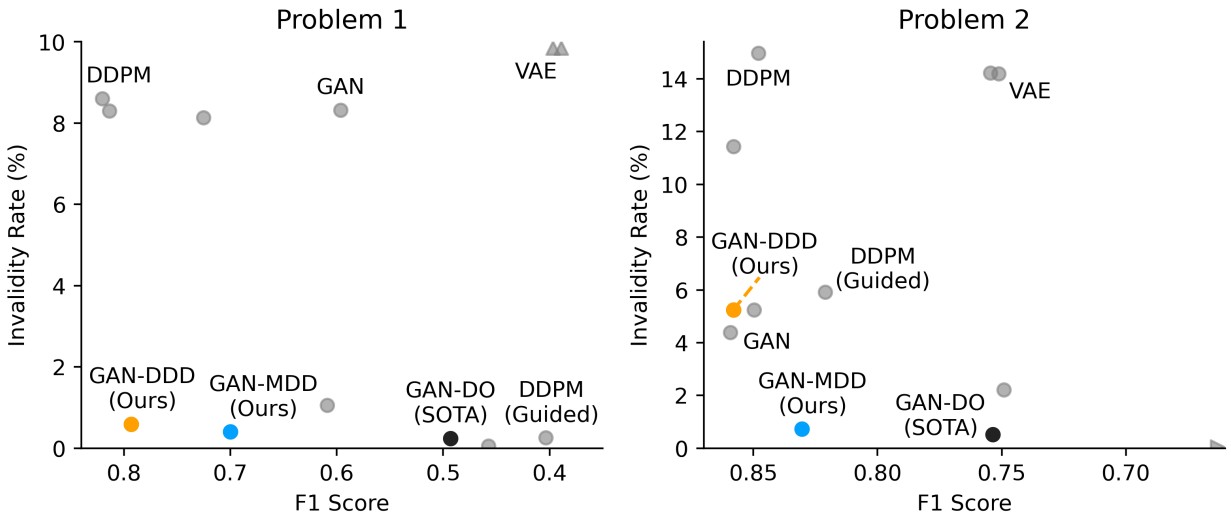

**Figure 24:** Comparison of GAN-DDD's F1 scores (↑) and invalidity rates (↓) to baselines on `Problem 1` (left) and `Problem 2` (right). Gray markers indicate scores for other models benchmarked in this paper, with a few selectively annotated. Mean scores over six instantiations are plotted. Scores closer to the bottom left are more optimal. Triangular markers indicate that the score lies off the plot in the indicated direction.

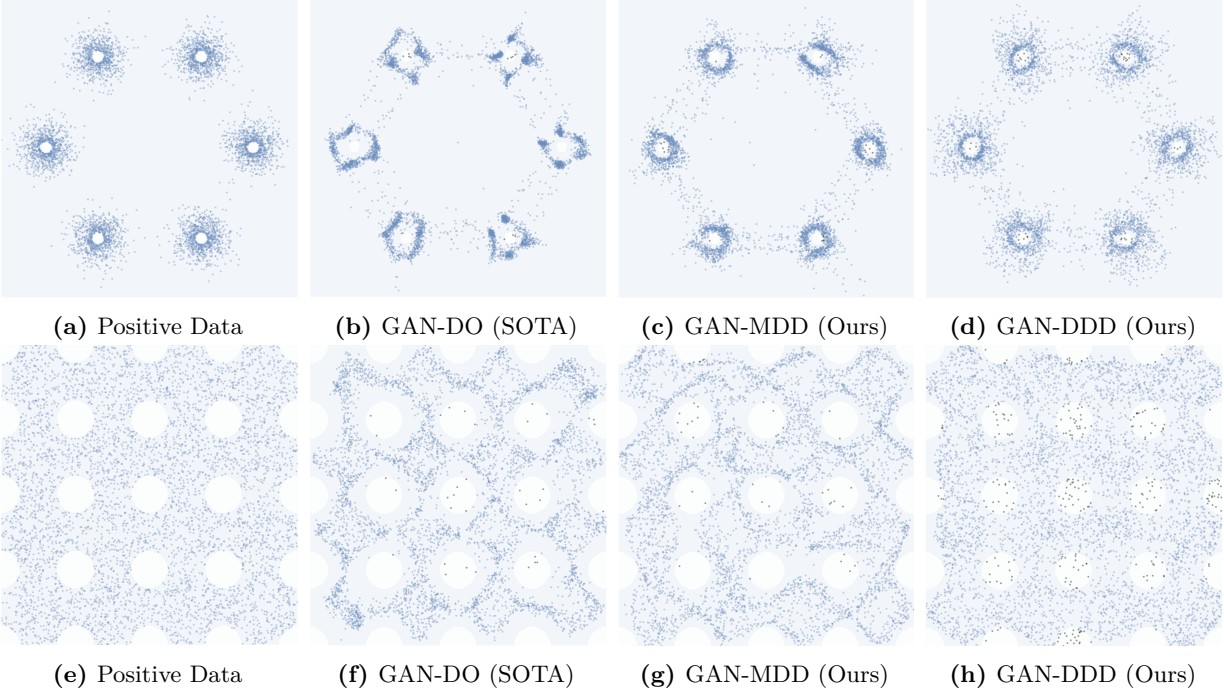

**(a)** Positive Data   **(b)** GAN-DO (SOTA)   **(c)** GAN-MDD (Ours)   **(d)** GAN-DDD (Ours)

**(e)** Positive Data   **(f)** GAN-DO (SOTA)   **(g)** GAN-MDD (Ours)   **(h)** GAN-DDD (Ours)

**Figure 25:** Comparison of GAN-DDD's generated distributions with those of GAN-DO and GAN-MDD on `Problem 1` and `Problem 2`.

# F  Details on Datasets, Models, and Training

## F.1  2D Experiments

### F.1.1  Dataset Details

`Problem 1`: Datapoints are randomly sampled from one of six modes, each of which is a 2D Gaussian. Distribution centers are spaced at an equal radius. Points in close proximity to the center of any distribution are labeled as negatives, while others are labeled as positives. Sampling is performed until 5K positive samples and 5K negative samples are acquired, and any excess in the oversampled class is discarded.

`Problem 2`: Datapoints are uniformly sampled. A square grid of 'centerpoints' is overlaid over the distribution. Datapoints within a specified proximity to a 'centerpoint' are considered negative, while all others are labeled as positive. Sampling is performed until 5K positive samples and 5K negative samples are acquired, and any excess in the oversampled class is discarded.

### F.1.2  Training Details

All tested networks (encoder, decoder, generator, discriminator, DDPM noise model, auxiliary discriminator) are deep networks with one hidden layer of 400 neurons and ReLU activations. A batch size of 256 is used throughout. Models are trained using the Adam optimizer (Kingma & Ba, 2014) with a learning rate $3 \cdot 10^{-4}$. Models are trained for 10000 epochs. The noise dimension for the GAN and latent dimension for the VAE are set at 8. Diversity weights are set at 0.1.

## F.2  Engineering Experiments

### F.2.1  Dataset Details

Several of the engineering datasets were compiled and described in Yu et al. (2024). For all datasets in this section, optimization objectives are not utilized.

**Ashby Chart**: Taken from (Jetton et al., 2023), this problem explores physically feasible combinations of material properties, according to known physical materials from an Ashby chart. The constraint function combines an analytical constraint and a lookup from an Ashby chart. Material properties considered are density, yield strength, and Young's modulus. Material classes included are foams, natural materials, polymers, composites, ceramics, and metals. 1K positive samples and 1K negative samples are selected using uniform random sampling.

**Bike Frame**: The FRAMED dataset (Regenwetter et al., 2022b) is comprised of 4292 in-distribution (positive) human-designed bicycle frame models. FRAMED also contains 3242 constraint-violating (negative) designs, some of which were human-designed and some of which were synthesized by generative models. FRAMED also contains 10095 generative model-synthesized valid designs that are not assumed to be in-distribution and are thus unused in this benchmark. Constraints consist of a set of empirical geometric checks and a black-box 3D reconstruction check. Constraints are unified using an all-or-nothing approach. Validity scores on this dataset are only evaluated using empirical checks.

**Cantilever Beam**: This problem considers the design of a five-component stepped cantilever beam. The thickness and height of each of the five components are the design variables, while the lengths of each component are given (fixed). Taken from (Gandomi & Yang, 2011), this problem has numerous geometric constraints and an overall constraint limiting the total deflection allowed by the design under a simple concentrated load at the tip of the beam. 1K positive samples and 1K negative samples are selected using uniform random sampling.

**Car Impact**: This problem quantifies the performance of a car design under a side impact scenario based on European Enhanced Vehicle-Safety Committee (EEVC) procedures (Gandomi et al., 2011). The car chassis is represented by 11 design parameters. Critical deflection, load, and velocity thresholds are specified for various components of a crash dummy, constituting 10 constraints. 1K positive samples and 1K negative samples are selected using uniform random sampling.

**Compression Spring**: This problem, taken from (Gandomi & Yang, 2011), centers around the design of a helical compression spring parameterized over coil diameter, wire diameter, and number of spring coils.

A constraint on free length and a constraint on displacement under a compressive load are specified. 1K positive samples and 1K negative samples are selected using uniform random sampling.

**Concrete Beam**: Taken from (Gandomi & Yang, 2011), this problem centers around the design of a simply supported concrete beam under a distributed load case. The beam is parameterized using a cross sectional area, base length, and height and is subject to a safety requirement indicated in the American Concrete Institute (ACI) 319-77 code. 1K positive samples and 1K negative samples are selected using uniform random sampling.

**Gearbox**: This gearbox (speed-reducer) design problem, taken from (Gandomi & Yang, 2011) features 7 parameters describing key geometric components like shaft diameters, number of teeth on gears, and face width of gears. Nine constraints are given, spanning considerations like bending stress on gear teeth, transverse stress and deflection on shafts, and surface stresses. 1K positive samples and 1K negative samples are selected using uniform random sampling.

**Heat Exchanger**: This problem, sourced from (Yang & Gandomi, 2012) considers the design of a heat exchanger involving eight design parameters and six constraints focused on geometric validity. 1K positive samples and 1K negative samples are selected using uniform random sampling.

**Pressure Vessel**: This cylindrical pressure vessel design problem is taken from (Gandomi & Yang, 2011). The pressure vessel is parametrized according to four parameters, namely the cylinder thickness, spherical head thickness, inner radius, and cylinder length. Four geometric and structural constraints are specified in accordance with American Society of Mechanical Engineers (ASME) design codes. 1K positive samples and 1K negative samples are selected using uniform random sampling.

**Ship Hull**: The SHIPD Dataset (Bagazinski & Ahmed, 2023) is comprised of 30k valid (positive) ship hull designs and 20k invalid (negative) ship hull designs. The SHIPD dataset includes numerous constraints spanning geometric rules and functional performance targets, focusing on various types of hydrodynamic performance.

**Three-Bar Truss**: Taken from (Yang & Gandomi, 2012), this truss design problem considers the design of a three-beam truss parameterized by the length of two of the beams (symmetry specifies the length of the third). The system is subject to one geometric constraint and two maximum stress constraints. 1K positive samples and 1K negative samples are selected using uniform random sampling.

**Welded Beam**: Taken from (Gandomi & Yang, 2011), this problem concerns a cantilever beam welded to a flat surface under a simple concentrated load at the tip of the beam. The beam is parametrized using a weld thickness, welded joint length, beam thickness, and beam width. Five structural constraints are given, specifying a maximum shear stress, bending stress, buckling load, and deflection, as well as a geometric constraint on the beam. 1K positive samples and 1K negative samples are selected using uniform random sampling.

### F.2.2 Training Details

The generator and discriminator are deep networks with one hidden layer of 400 neurons and ReLU activations. A batch size of 256 is used throughout. Models are trained using the Adam optimizer with a learning rate $3 \cdot 10^{-4}$. Models are trained for 10000 epochs. The noise dimension for the GAN and latent dimension for the VAE are set at 8. Diversity weights are tuned by selecting the highest score from 10 increments tested.

### F.3 Topology Optimization Experiments

### F.3.1 Dataset Details

The GAN was trained exclusively on 32436 valid (connected) topologies generated through iterative optimization (SIMP) (Bendsøe & Kikuchi, 1988). The GAN-MDD and GAN-DO models are trained on a medley of disconnected topologies generated by iterative optimization (2564), and either procedurally-generated synthetic topologies (35000) or GAN-generated disconnected topologies (92307). Synthetic topologies were sourced directly from the classification dataset of (Mazé & Ahmed, 2023). The GAN used to generate disconnected topologies for rejection was the exact GAN benchmarked in the paper (the first instantiation of six). Topologies were checked for continuity and rejected samples were added to the negative dataset. All

positive and negative data were augmented sevenfold using horizontal and vertical flips, as well as quarter-turn rotations, prior to training.

### F.3.2 Training Details

The model architectures of the GAN, GAN-DO and GAN-MDD are identical except for the final output dimension of the discriminator. Both the generator and discriminator are simple 5-layer convolutional neural networks. The generator has 3.6M parameters, while the discriminator has 2.8M parameters. For more architectural details, we refer the reader to the codebase. The latent dimension is 100, batch size is 128, and learning rate for both models is $3 \cdot 10^{-4}$, using the Adam optimizer.

