# OpenReview forum: "Constraining Generative Models for Engineering Design with Negative Data"
_TMLR — Accepted by TMLR_

### Review · Reviewer_SoqL · 2024-07-01

**Summary Of Contributions:**

- contributions
	- Develops two approaches to incorporating negative data into generative models (specifically GANs):
		- multi-class model: Trains a multi-class classifier to distinguish positive, negative, and generated samples as an auxiliary GAN loss.
		- double discriminator: Train two discriminators, one for positive samples and one for negative samples. A weighted combination of these is used in the GAN training loss.
	- Evaluates several negative data generative model variants (and baseline models that do not use negative samples) across toy examples, an engineering task suite, a block stacking task, and topology optimization for engineering design.
	- Evaluates some of the scaling properties obtained by adding negative samples.
- new knowledge
	- GANs outperform DDPM (representing the class of diffusion models) on engineering tasks with hard constraints.
	- Modest amounts of negative samples can substantially improve model performance in terms of satisfying hard constraints (at least for some tasks).
	- Negative samples can vary in how much they improve performance based on the form of the negative samples (being "easy" or "hard"). Specifically contrasts samples generated from heuristically generated synthetic data and samples generated by rejection sampling from a baseline generative model.

**Audience:**

Yes

**Broader Impact Concerns:**

Broader impact is not directly addressed in an explicit Broader Impact Statement section. It does not seem necessary as the goal of the work is enabling better control over the outputs of generative models.

**Claims And Evidence:**

No

**Requested Changes:**

# critical
- In the main text tables please include the following:
	- Estimates of statistical variability for all results reported in tables.
	- Estimates of effect sizes and statistical significance for any differences claimed. For example, if the claim is GAN-MC outperforms GAN, include an estimate of the magnitude of improvement.
- Include GAN-DO, GAN-DD, GAN-MC in all mainline reported results.
	- These are the contributed methods (and strong baseline in the case of GAN-DO) that are claimed contributions, and thus merit comprehensive evaluation.

# strengthen
- Ideally also include DDPM in all the evaluations. The claim that DDPM is inferior is very interesting and merits more comprehensive evaluation and summarization.
- (less critical) The potential of negative samples to improve generative models is fascinating. I wonder about two scaling studies:
	- 1) Extending the results reported in Table 2 to include other methods (particularly DDPM given it's popularity as a model elsewhere)
	- 2) Extending the scaling range to cases where models train on more negative samples than positive. In at least some engineering domains it is easy to generate negative samples but positive samples are scarce. How well do the sample efficiency results extend to the regime of far more negative than positive samples?
		- The GAN-DD results from Table 2 are quite promising in this regard. 4x as many positive samples (from 1K) only yields a ~0.1 percentage point improvement (if any) for models with 4K or 16K negative samples. If possible this same scaling would be interesting to apply to the topology optimization problem, given the ability to generate negative samples there.


Other notes to help strengthen the submission:
- Some experiments would benefit from matching the number of training samples used in total (positive and negative). some comparisons are somewhat "unfair" in providing the model access to additional data (as negative samples) that are not used by vanilla baselines
- (reiterating above) A general remark on the tables: Please report the number of trials for each table (ideally in the caption). The tables should include reports of statistical variability (often included in the appendix) and some statistical test of differences for any differences among models being claimed. This remark applies to all the tables provided and claims about differences in the main text.
- Table 1
	- It would be interesting to see these results but training the vanilla models (GAN, VAE, DDPM) with additional positive samples to match the total number of samples given to the negative models. For example, providing 20k positive samples to the vanilla models to make the 10k positive and 10k negative. The idea is to be somewhat more "fair" in granting access to the same amount of data to learn from, where the only difference is whether the samples come from the negative distribution.
- Table 2
	- I find it interesting that 4K positive samples often outperforms 16K for both problems and most model (except GAN-DD 1K). Any idea why this might be?
	- This table should include GAN-MC results for the same scaling test.
	- It would be interesting to see DDPM performance on this task: do they scale more poorly than GANs? Could the problem be that DDPMs require more data to start being effective (hence the negative main result in the experiments so far)?
	- This table should at least include GAN-MC results.
- Table 3
	- This table should include GAN-MC results.
- Table 4
	- This table should include GAN-DD results.
- Table 5
	- This table should include GAN-DO and GAN-DD for comparison.
	- How many pixels are in the output images? It is hard to interpret a score of 0.29 pixels without that information.
- Tables 14-19
	- These tables include "GAN-D2" and "GAN-RM", but I could not find either defined in the text.
	- These tables lack reporting of variability over multiple trials (in addition to lacking statistical tests of differences for the claims being made in the text).
- Section 5.1
	- "Figure 5 plots the datasets and the generated distributions of several select models (vanilla GAN and the two NDGMs we propose)"
		- "Figure 5" should be "Figure 2"

**Strengths And Weaknesses:**

# strengths
- Provides strong evidence for the importance of negative data to a breadth of engineering tasks. The breadth of evidence is compelling and there are a healthy variety of alternative models tested.
- Contributes two conceptually straightforward GAN extensions with reasonably good empirical performance.


# weaknesses
- Lack of reporting statistical variability in the main text (supplement includes some)
	- There is no reporting of statistical tests of differences among models when claims are made about which model or models are better (worse) for a given task. The appendix tables include measures of statistical variation, so the text should include empirical estimates of effect sizes (and statistical significance) when comparing models. This will strengthen the claims about differences among models.
- The two contributed models are not included in all of the evaluations.
	- GAN-MC and GAN-DD-a (and possibly GAN-DD-b) should be included in all tables in the main text as they are the main technical contributions of the paper.
	- It would help to include the DDPM model variants in all results as well. Or a table clearly summarizing their performance. The claim that diffusion models are inferior on these tasks is interesting and provocative, so providing strong evidence of the consistency of this trend (or specificity to particular evaluations) will be valuable. The text also lacks a single easy reference summary for readers to assess the evidence in favor the the inferiority of diffusion models across the many evaluations.
- (minor) The benchmarks do not include any sense of what "good" performance is.
	- Is there a human baseline or other baseline to include to compare these generative methods to?
	- A "gold standard" accepted by engineers in the respective tasks?
	- This would strengthen the ecological validity of these tasks.

---

> ### Author Response · Authors · 2024-08-25
>
> **General Notes**: We thank the reviewer for the positive feedback and for identifying the importance of negative data in engineering tasks. Likewise, we thank the reviewer for raising important concerns with statistical testing of the results. Thanks to the reviewer's feedback, we added more experiments to establish statistical confidence in many of the claims. In these results, we have included the strong (GAN-DO) and weak (GAN) baselines and directly compared our method (GAN-MC) to confidently establish superiority in many of the metrics and problems. Since the reviewer's comments affect all the results in the paper, we thought it pragmatic to summarize the updated results, rather than address each critique point-by-point.
>
> ***Figures and Tables are included in a supplementary pdf that can be found at this [link](https://anonymous.4open.science/r/ndgm_rebuttal-E4B1/NDGM_TMLR-1.pdf)***
>
> Prompted by feedback from Reviewer W5Nm, we have decided to incorporate diversity as a core component of our contribution and in turn move GAN-DD to the appendix to simplify. We therefore test the diversity-augmented version of GAN-MC in the results included in this rebuttal (and plan to update the paper's results accordingly).
>
> We have remade many of the results, in many cases including more tests. In the new results, we have included standard deviations of scores and incorporated statistical testing to evaluate the explicit claims made in the paper. Statistical testing was performed throughout using 2-sample t-tests. Our updated results support many of the claims in the paper. For those that are not supported with statistical confidence, we plan to adjust our claims in accordance with our findings.
>
> Due to the quantity of experiments that had to be run in a short amount of time and our desire to establish statistical significance, we made a few adjustments to our code to accelerate training. One of these modifications was combining gradients of disparate loss terms into a single optimizer step, which both accelerated training and improved model performance. For fairness, this modification was made both to our proposed methods and to baselines tested. Due to this change, slight differences between previous results and new results are possible. Thanks to these enhancements, we were able to run a variety of tests and establish statistical confidence in many of the paper's claims. However, we were unable to prepare all of the requested adjustments to the paper, but plan to complete the suggested modifications for any future drafts.
>
> We appreciate the reviewers interest in the relative performance of GAN models and diffusion models, and acknowledge the current excitement for Diffusion in many ML communities. It was not our intent to make the value comparison between diffusion models and GANs a prominent aspect of the paper, as we feel that this detracts from the intended focus and the methods.
>
> ***2D Problems***: We benchmark GAN-MC GAN-DO and GAN over 7 instantiations on the 2D problems to establish greater statistical confidence in the findings. For this experiment, we also slightly adjusted the bandwidth parameter and cluster count parameters of the MMD and F1 scores, respectively, to better emphasize fine distributional details and make differences between models more prominent. Results can be seen in Table 1 in the attached PDF. Both GAN-DO and GAN-MC significantly outperform GAN in validity, but GAN-DO also significantly outperforms GAN-MC in Problem 2. However, GAN-MC performs significantly better than GAN-DO and GAN in both statistical similarity scores in both problems.
>
> We plan to extend these results statistical testing and the conclusions presented to the dozen other baselines currently in the main paper.
>
> ***Dataset Size:***
> We have remade the dataset size study with GAN-MC, rather than GAN-DD in Table 2. We have also included standard deviations of scores. We also reran the GAN model, which previously displayed unexplained performance degradation with larger datasets. GAN-MC with 1K positive and 1K negative outperforms GAN with 16K positive with $p<0.05$ in both test problems, representing a significant performance improvement with 1/8 the data.
>
> Continued in next comment...

---

> ### Author Response · Authors · 2024-08-25
>
> Continued from previous...
>
> ***Engineering:***
> Thanks to the reviewer's critique, we identified that many of our results and our claims on the engineering experiments were not statistically significant. To establish conclusions with more confidence, we felt the need to rerun the engineering experiments with more trials (we ended up testing 7 instantiations of each model rather than 3 from the original paper). Using these new results, we identified statistically significant conclusions.
>
> First, we consider constraint violation (invalidity) scores (see Table 3). In summary, GAN-MC and GAN-DO both dominate GAN in pure constraint satisfaction, achieving statistically significant performance gaps over the GAN in the majority of test problems. GAN-MC achieves the highest mean score in four problems while GAN-DO achieves the highest mean score in eight. GAN-DO appears to have a slight edge over GAN-MC in a direct comparison, significantly outperforming GAN-MC in 6 problems, while GAN-MC only significantly outperforms in 1 problem.
>
>
> When examining distributional similarity scores (Table 4), GAN-DO can sometimes be seen to trail the GAN. In four problems, it achieves significantly worse F1 score compared to the GAN and only has the highest mean score in one problem. In comparison, GAN-DO achieves significantly better scores in three problems and significantly worse scores in three problems and can be roughly summarized as performing on-par with the GAN (GAN-MC has the highest mean score in five problems to GAN's six). A direct comparison of GAN-MC to GAN-DO reveals that GAN-MC significantly outperforms GAN-DO in statistical similarity in six problems while the reverse is only true in two.
>
> Broadly speaking, while GAN-DO is often the strongest model in pure constraint satisfaction, this can come at the cost of distributional similarity. In contrast GAN-MC still achieves excellent constraint satisfaction, while generally achieving stronger distribution-matching performance compared to GAN-DO.
>
> ***Block Stacking***: We unfortunately did not have time to benchmark GAN-MC on the block stacking problem and will either add this benchmark in a future draft of the paper or remove the block stacking example if the other results take too much space (or move it to the appendix).
>
>
> ***Topology Optimization Experiments***: Following the reviewer's suggestion, we have benchmarked GAN-DO. Unfortunately, these TO models are a bit larger (4M parameters) and take nearly a day to train on our available hardware. Therefore, we were unable to benchmark these models numerous times and establish statistical confidence in the quantitative results. The models have only been run 1-2 times, however we will continue to test instantiations until we can make quantitative claims with confidence. Regardless, we still find these experiments to be  insightful from a qualitative standpoint, as it's clearly visible that GAN-DO suffers from lack of diversity despite its relatively strong constraint satisfaction scores.
>
> Quantitative results are shown in Table 5. GAN-DO seems to score better than GAN-MC by a modest margin when training using rejected negative data, but underperforms when using synthetic negative data and rejected data. Though the data is currently not sufficient to establish relationships between models with statistical confidence, we hope to do so soon.
> Quantitative results are shown in Table 5. GAN-DO seems to score better than GAN-MC by a modest margin when training using rejected negative data, but underperforms when using synthetic negative data and rejected data. Though the data is currently not sufficient to establish relationships between models with statistical confidence, we hope to do so soon.
>
> Though GAN-DO appears a competitive choice based on pure constraint satisfaction scores, qualitative examination reveals that GAN-DO achieves this success largely due to a collapsed posterior distribution centered on just a few data modes. As shown in Figure 1, many of the generated topologies share the same general structure and some designs are nearly indistinguishable, suggesting a serious lack of diversity. In fact, the overwhelming majority of designs seem to fall within one of five groups (Seen in Figure 2), GAN-MC does not share this diversity issue (See Figure 3, copied here from the original paper's appendix for comparison). Though a few topologies have similar structure, they are fairly diverse.

---

> > ### Comment · Reviewer_SoqL · 2024-08-27
> >
> > Thank you for the extensive new experiments and analysis!
> >
> > The many changes to improve training efficiency and run tests will be welcome additions to the paper.
> >
> > Currently it appears that the narrative is somewhat shifted: GAN-MC is a strong candidate for tasks where both hard constraints and diversity matter, while GAN-DO is a strong candidate when hard constraints are the primary concern.
> >
> > My read of the current results is that leveraging negative data can substantially improve models meeting hard constraints. We have two strong approaches for this task that differ in the diversity of samples they produce. The problem is sufficiently hard and valuable to merit additional investigation. The proposed method (GAN-MC) is a meaningful first step to improving scaling behavior and sample diversity in at least some cases. There are many open problems in terms of the right data (mix) to use, how much data helps, and how to preserve diversity.
> >
> > It would help to see an overview of the main points of the narrative in light of the statistical tests and current results.
> >
> >
> > > RE: results reporting
> >
> > The yellow vs green checkmark annotation is confusing and not color-blind friendly. I would recommend using different symbols for when GAN-MC is (statistically significantly) superior vs GAN-DO is superior. Including the totals as narratively reported above (in 6 cases GAN-MC beats GAN-DO for ...) as part of the tables would also help.

---

> > > ### Author Response · Authors · 2024-08-28
> > >
> > > We thank the reviewer for the further comments, which are once again very insightful. We have a few responses.
> > >
> > > ***Reviewer Comment***: Currently it appears that the narrative is somewhat shifted: GAN-MC is a strong candidate for tasks where both hard constraints and diversity matter, while GAN-DO is a strong candidate when hard constraints are the primary concern.
> > >
> > > ***Author Response***: We agree that the intended narrative of the paper has shifted. We note that achieving perfect validity is quite trivial if a model collapses in a valid region of the space. Realizing this, we felt the need to focus not only on pure constraint satisfaction but also distributional similarity. We see GAN-MC as the no-compromise solution that does not underperform vanilla models in distributional similarity, yet achieves significantly better constraint satisfaction. GAN-DO often underperforms in distributional similarity (not just diversity), but could be a strong candidate if hard constraints are the primary concern. However, given that a user is performing generative modeling in the first place (rather than direct design optimization, for example), it stand to reason that they would care about distribution matching to a reasonable extent, which may make GAN-DO less strong of a choice.
> > >
> > > ***Reviewer Comment***: My read of the current results is that leveraging negative data can substantially improve models meeting hard constraints. We have two strong approaches for this task that differ in the diversity of samples they produce. The problem is sufficiently hard and valuable to merit additional investigation. The proposed method (GAN-MC) is a meaningful first step to improving scaling behavior and sample diversity in at least some cases.
> > >
> > > ***Author Comment***: The reviewer's read of the updated results closely mirrors our vision for reframing the results and takeaways in an future draft of the paper.
> > >
> > > ***Reviewer Comment***: It would help to see an overview of the main points of the narrative in light of the statistical tests and current results.
> > >
> > > ***Author Response***:
> > >
> > > To summarize the (adjusted) key contributions:
> > > - We perform an extensive benchmark of NDGMs on a variety of engineering-related and other problems showing widespread improvements in constraint-satisfaction and data-efficency.
> > > - We propose a novel training formulation combining the proposed multiclass discriminator adversarial model with a diversity-based loss.
> > > - We demonstrate that our model performs significantly better in constraint satisfaction than vanilla models in most problems and significantly better in distributional similarity than strong NDGM baselines in many problems.
> > >
> > > To summarize key insights from the paper that we feel readers can gain:
> > > - NDGMs are very potent in engineering design problems with constraints. We encourage their use when negative data is available or can be easily collected. Use of NDGMs in engineering design is almost unheard of and the potential improvement is very significant.
> > > - NDGMs can be massively more data-efficient over vanilla models.
> > > - The many options for NDGMs give users some choice, but GAN-DO and GAN-MC stand out are strong baselines. While GAN-DO often achieves very strong constraint satisfaction, it may do so with a concerning loss of sample diversity, leading it to underform GAN-MC in distribution-matching. If constraint satisfaction is of singular importance, GAN-DO is likely the best choice, but GAN-MC is consistently a strong and balanced choice.
> > >
> > >
> > > ***Reviewer Comment***: There are many open problems in terms of the right data (mix) to use, how much data helps, and how to preserve diversity.
> > >
> > > ***Author Response***: We agree that there are open questions in data mixture, data quantity, and diversity preservation. We feel that the paper's dataset size study is a meaningful step towards understanding data mixture and quantity, while the ablation studies with diversity loss in the appendix are a step towards understanding diversity in NDGMs (we will update these ablations in the next draft, perhaps including diversity scores). We feel that further exploration in these subjects would be difficult to include given the space constraints of the paper and may dilute the focus of the work. Hence feel that they would be better addressed in dedicated follow-up work.
> > >
> > > ***Reviewer Comment:*** The yellow vs green checkmark annotation is confusing and not color-blind friendly. I would recommend using different symbols for when GAN-MC is (statistically significantly) superior vs GAN-DO is superior. Including the totals as narratively reported above (in 6 cases GAN-MC beats GAN-DO for ...) as part of the tables would also help.
> > >
> > > ***Author Response***: We thank the reviewer for the suggestions to improve the readability and accessibility of the new tables. We agree with the recommendations and will gladly incorporate them in the next draft.

---

### Review · Reviewer_W5Nm · 2024-08-01

**Summary Of Contributions:**

For generative modeling with constrains, the paper proposed two approaches to guide models toward constraint-satisfying outputs using negative data. With extensive benchmarks, the NDGM models outperform against baseline models.

**Audience:**

Yes

**Claims And Evidence:**

Yes

**Requested Changes:**

It will be great to add the sample diversity discussion in the main paper. So that readers can understand the tradeoff of new approaches easier.

**Strengths And Weaknesses:**

**Strength**
1) The NDGM Formulations are quite easy to understand, and the authors provided extensive benchmarks.
2) The experiments showed improved constraint satisfaction rates and improved data efficiency.

**Weakness**
1) This paper mainly focuses on GAN models instead of diffusion models. It's interesting to compare with other diffusion models.
2) In the main paper, there is no discussion about the sampling diversity.

---

> ### Author Response · Authors · 2024-08-25
>
> **General Notes**: The authors thank the reviewer for the feedback. We appreciate that the reviewer found the formulation easy to understand and the experiments compelling.
>
> ***Reviewer critique***: "This paper mainly focuses on GAN models instead of diffusion models. It's interesting to compare with other diffusion models."
>
> ***Response***: We acknowledge the current excitement among diffusion models across many research communities. We note that in many engineering design applications, GANs remain state-of-the-art. It was not our intent to make any sort of value comparison between diffusion models and GANs. We simply benchmarked a few DDPM variants for completeness. In a paper with such a wide array of benchmarks, we feel that benchmarking more diffusion model variants and comparing their performance may distract from the core message and contributions of the paper.
>
> ***Reviewer critique***: "In the main paper, there is no discussion about the sampling diversity."
> ***Requested change***: "It will be great to add the sample diversity discussion in the main paper. So that readers can understand the tradeoff of new approaches easier."
>
> ***Response***: The authors thank the reviewer for this feedback. Noting that diversity makes a significant contribution to the distributional similarity performance of our NDGM formulation, we have reconsidered our decision to incorporate this as appendix material. Instead, we would like to bring it into the main paper as one of the core contributions. Since this adds some complexity to the core method of the work, we thought it wise to centralize the contributions around one formulation, rather than proposing two in the main paper. Therefore, we plan to move discussion of GAN-DD to the appendix and propose GAN-MC + Diversity as the main methodology contribution of the paper (adjusting the experiments as such). Importantly, while diversity significantly improves distributional similarity, it slightly damages validity. Thus, our model does not always outperform baselines like GAN-DO in validity anymore, but instead almost always does so in distributional similarity. Therefore, to illustrate the benefits of the approach, we will add distributional similarity scores to some of the sections that previously only scored validity.
>
> Many experiments have been rerun at Reviewer SoqL's suggestion. Updated results can be found in our response to their review. Results labeled GAN-MC in these results incorporate the diversity-based loss.

---

### Review · Reviewer_RJhg · 2024-08-12

**Summary Of Contributions:**

This work introduces two new Negative-Data Generative Models (NDGMs) that match or surpass state-of-the-art methods. The proposed NDGMs can significantly outperform vanilla models using less data. Meanwhile, to evaluation it, the paper also includes an extensive benchmark set across many synthetic problems and engineering tasks.

**Audience:**

Yes

**Broader Impact Concerns:**

I'm not sure how much of an impact this work will have on the field of Engineering Design.I am not an expert in the field of Engineering Design.

**Claims And Evidence:**

No

**Requested Changes:**

No change request.

**Strengths And Weaknesses:**

Strengths:
1. The proposed NDGMs significantly enhance constraint satisfaction across a variety of synthetic and challenging engineering problems.
2. NDGMs  achieve superior performance using much less data compared to other models, highlighting their data efficiency.
3. The paper includes a comprehensive benchmark, providing a robust evaluation framework for NDGMs.

Weaknesses:
1. The use of dual and multi-class discriminators may result in higher computational costs and longer training times, particularly for large-scale datasets.
2. Although the methods are effective, the proposed multi-category discriminator and dual-discriminator methods are not technically significant innovations. However, the results are good.
3. A simple application of GAN in the field of Engineering Design. Whether it is inspiring for other fields is uncertain.

---

> ### Author Response · Authors · 2024-08-25
>
> **General Notes**: We thank the reviewer for the positive feedback and appreciate that the reviewer found the results compelling. We have some comments on the reviewer's key critiques below.
>
> ***Reviewer Critique***: "The use of dual and multi-class discriminators may result in higher computational costs and longer training times, particularly for large-scale datasets."
>
> ***Response***: Thanks for this critique. It is true that the dual discriminators increases training cost. However, due to feedback from Reviewer W5Nm, we have decided to centralize the core contribution around the multiclass version augmented with diversity loss. The multiclass discriminator adds negligible (within 1\%) cost. If augmented with diversity loss, this loss adds computational cost to the training. However, as demonstrated in the paper, even without diversity, the proposed GAN-MC yields significant benefits both in terms of validity and statistical similarity at negligible cost.
>
> ***Reviewer Critique***: "Although the methods are effective, the proposed multi-category discriminator and dual-discriminator methods are not technically significant innovations. However, the results are good."
>
> ***Response***: We acknowledge that the proposed methods are relatively simple, but we feel that this simplicity is a strength rather than a limitation. Negative data is readily available in numerous engineering design problems, yet has not been utilized because the methods were not yet pioneered. Therefore, we feel that our method, although simple, is a technically significant innovation, particularly with the addition of the diversity-based training objective as one of the core technical contributions.
>
> ***Reviewer Critique***: "A simple application of GAN in the field of Engineering Design. Whether it is inspiring for other fields is uncertain."
>
> ***Response***: Although we focus on engineering and design applications, we expect negative data to have broader success in many generative fields involving hard constraints. We also feel that engineering design spans many important and societally impactful technologies (material and drug discovery, structural engineering, product design, etc.). These problems feature important hard constraints. We feel that establishing negative data models as a potent framework for learning constraints in engineering design is an important stepping stone. We would be excited to test our methods on well-known high-impact benchmarks and problems in the future.

---

### Decision · Action_Editor_1cUU · 2024-10-23

**Recommendation:** Accept with minor revision

**Comment:**

All reviewers agree on the simplicity and superior performance of the proposed method, as evidenced by extensive experiments. The concerns raised regarding computational cost, sample diversity, and statistical variability have been thoroughly addressed by the authors. Considering the strong performance in the engineering task domain, I support the acceptance of this paper.

Please prepare the final version of the manuscript, ensuring that you adequately incorporate the reviewers' comments and your responses to them.

**Audience:**

This paper could be of interest to parts of the TMLR audience.

**Claims And Evidence:**

The authors developed a Negative-Data Generative Model (GAN-MC with diversity techniques) to guide generative models toward producing constraint-satisfying outputs using negative data. Through extensive evaluations on an engineering task suite, a block stacking task, and topology optimization for engineering design, they empirically demonstrated the superiority of their proposed method in terms of distributional similarity when compared to existing methods, including DDPM.

---

> ### Author Response · Authors · 2024-10-24
>
> We would like to express our gratitude to the reviewers for their valuable feedback and to the AE for their coordination efforts and decision. We look forward to preparing the revised submission!